# SuCo: Sufficiency-guided Continuous Adaptive Reasoning

**Jiahao Wang** [1]  **Bingyu Liang** [1]  **Chenhao Hu** [1]  **Longhui Zhang** [✉][1]  **Xuebo Liu** [1]  **Min Zhang** [1]  **Jing Li** [✉][1]  **Xuelong Li** [✉][2]

## Abstract

Despite remarkable performance on complex tasks, Large Reasoning Models (LRMs) often generate excessively long Chain-of-Thoughts (CoT), inflating computational costs even for simple queries. Existing efforts to mitigate this inefficiency typically rely on discrete reasoning modes or fixed budget tiers, lacking a principled criterion of when reasoning is sufficient. In this work, we introduce *Minimal Sufficient CoT* (MSC), defined as the shortest prefix of a CoT trajectory which is adequate for producing the correct answer. We empirically show that MSC not only reduces reasoning tokens, but also improves accuracy across difficulty levels. Building on MSC, we propose *Sufficiency-guided Continuous Adaptive Reasoning* (SuCo), a two-stage training framework for autonomous reasoning control along a continuous spectrum. In stage I, *MSC-Aligned Fine-Tuning* (MFT) constructs MSC data using problem-adaptive sufficiency thresholds that naturally scale with question difficulty, then fine-tunes the model to internalize concise yet sufficient reasoning patterns. In stage II, *Sufficiency-Aware Policy Optimization* (SAPO) further optimizes the model through reinforcement learning with dynamic complexity tracking and sufficiency-aware rewards that penalize both over- and under-thinking. Extensive experiments across mathematics, code, and science benchmarks show that SuCo consistently achieves improvements in both accuracy and reasoning efficiency.

[1]Harbin Institute of Technology (Shenzhen), Guangdong, China. [2]TeleAI of China Telecom, China. Correspondence to: Longhui Zhang <longhuizhang97@gmail.com>, Jing Li <jingli.phd@hotmail.com>, Xuelong Li <xuelong_li@ieee.org.>.

*Proceedings of the 43rd International Conference on Machine Learning*, Seoul, South Korea. PMLR 306, 2026. Copyright 2026 by the author(s).

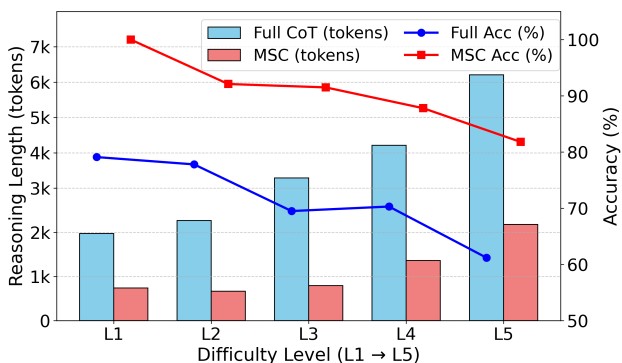

*Figure 1.* MSC vs. Full CoT on Qwen3-8B across MATH difficulty levels. **Left axis** (↓): reasoning tokens. **Right axis** (↑): accuracy. At each difficulty level, MSC achieves higher accuracy with significantly fewer tokens.

## 1. Introduction

Large Language Models (LLMs) have demonstrated impressive capabilities across a wide range of tasks (Zhao et al., 2023; Wang et al., 2025a; Zhang et al., 2025c), yet continue to struggle with complex problems requiring multi-step reasoning (Cobbe et al., 2021). To address this limitation, recent work has introduced *Large Reasoning Models* (LRMs), which explicitly generate intermediate reasoning steps via Chain-of-Thoughts (CoT) (Wei et al., 2022). By performing step-by-step logical thinking before arriving at final answers, LRMs such as DeepSeek-R1 (Guo et al., 2025) and OpenAI o1 (Jaech et al., 2024) achieve substantial gains over standard LLMs on challenging benchmarks (Hou et al., 2025b; Xu et al., 2025).

Despite these advances, current LRMs suffer from *redundant reasoning* (Sui et al., 2025). Even for simple queries, they tend to generate exhaustive reasoning chains, incurring substantial computational costs and inference latency (Aggarwal & Welleck, 2025). Such inefficiency limits practical deployment in real-time applications (e.g., online coding assistants (Jimenez et al., 2024)) and resource-constrained environments (e.g., edge devices (Zhang et al., 2024)).

To mitigate redundancy, recent studies have developed *Adaptive Large Reasoning Models* (ALRMs), which aim to adjust reasoning effort according to problem complexity (Sui

et al., 2025; Wu et al., 2025). These approaches can be broadly categorized into two paradigms. *User-controlled methods* require explicit prompts to select reasoning behaviors. For example, Qwen3 (Qwen Team, 2025) enables manual on/off switching, while GPT-OSS (OpenAI, 2025) provides multiple predefined reasoning strategies. In contrast, *model-driven methods* allow autonomous reasoning decisions. AdaCoT (Lou et al., 2025) employs external assessors, whereas LHRM (Jiang et al., 2025) assigns reasoning status based on domain labels. Despite their differences, existing ALRMs fundamentally rely on **discrete mode selection**. Reasoning effort is adjusted by switching among a finite set of manually specified options, rather than being calibrated in a continuous manner.

We posit that an ideal ALRM requires: (1) reasoning length scales with problem difficulty, (2) autonomous resource allocation without intervention, and (3) optimal performance with minimal reasoning. However, this raises a counterintuitive question: According to the test-time scaling laws (Snell et al., 2025; Brown et al., 2024), performance typically improves with more reasoning. **Can models actually perform better with less reasoning?**

We provide an affirmative answer by introducing **Minimal Sufficient CoT (MSC)** — the shortest reasoning prefix of a CoT trajectory that is sufficient to yield the correct answer. As illustrated in Figure 1, across all five difficulty levels of the MATH benchmark (Hendrycks et al., 2021b), MSC dramatically reduces reasoning tokens while consistently outperforming full CoT in accuracy. This reveals that rather than blindly scaling reasoning resources, test-time adaptation offers a more efficient solution.

Building on this insight, we propose **Su**fficiency-guided **Co**ntinuous Adaptive Reasoning (**SuCo**), a two-stage training framework enabling continuous reasoning control. Unlike prior discrete approaches that depend on external classifiers or predefined budget tiers, SuCo introduces problem-adaptive sufficiency thresholds that naturally adjust to question difficulty. In Stage I, *MSC-Aligned Fine-Tuning* (MFT) constructs an MSC dataset from full CoT trajectories, then performs supervised fine-tuning (SFT) to internalize concise yet sufficient reasoning patterns. In Stage II, *Sufficiency-Aware Policy Optimization* (SAPO) further trains the model to dynamically allocate reasoning effort through reinforcement learning (RL). Critically, SAPO maintains a dynamic complexity pool to track evolving reasoning distributions during training, and employs sufficiency-aware rewards that penalize both insufficient and excessive reasoning.

Extensive experiments are conducted across mathematics, code, and science domains at both 1.5B and 7B model scales. Results demonstrate that SuCo achieves superior accuracy with substantially fewer reasoning tokens, outperforming full CoT and ALRM baselines.

Our key contributions are summarized as follows:

- We formalize MSC, providing a principled sufficiency criterion revealing that models can achieve stronger performance with less reasoning.
- We propose SuCo, a two-stage training paradigm for continuous and autonomous reasoning control without discrete modes or external intervention.
- Comprehensive experiments spanning diverse domains demonstrate the effectiveness of our SuCo.

## 2. Related Work

**Large Reasoning Models.** Large Reasoning Models (LRMs) extend Large Language Models (LLMs) by explicitly generating intermediate reasoning steps via Chain-of-Thoughts (CoT), which has been shown to substantially improve performance on challenging multi-step tasks (Wei et al., 2022; Kojima et al., 2022). Building on this paradigm, recent LRMs such as DeepSeek-R1 (Guo et al., 2025), OpenAI o1 (Jaech et al., 2024), and Qwen3 (Qwen Team, 2025) further strengthen reasoning capabilities through large-scale supervised fine-tuning (SFT) on high-quality CoT data, often combined with reinforcement learning (RL) with curated rewards. Despite these advances, current LRMs frequently produce unnecessarily verbose reasoning even for trivial queries, incurring significant inference overhead and motivating the need for more efficient reasoning control.

**Adaptive Large Reasoning Models.** To mitigate reasoning redundancy, recent efforts have explored Adaptive Large Reasoning Models (ALRMs) that modulate reasoning length based on problem difficulty. Early approaches primarily focus on *binary triggering* of reasoning. AdaCoT (Lou et al., 2025) employs an external model to decide whether to activate CoT; AdaptThink (Zhang et al., 2025b) formulates reasoning activation as a constrained optimization problem; LHRMs (Jiang et al., 2025) assigns reasoning behaviors using coarse domain-level labels (e.g., math vs. chat). Beyond binary control, subsequent methods investigate *multi-mode reasoning*. SABER (Zhao et al., 2025) and ThinkDial (He et al., 2025) introduce multiple predefined reasoning strategies or budget tiers, selected via system prompts. Additional recent efforts have explored more fine-grained control. ThinkPrune (Hou et al., 2025a) applies reinforcement learning to prune long reasoning chains, while CyclicReflex (Fan et al., 2026) schedules reflection tokens cyclically to balance depth and efficiency. AlphaOne (Zhang et al., 2025a) explores dual-speed reasoning at test time, enabling models to adaptively think slow or fast. Complementary analyses have also highlighted the phenomena of *underthinking* (Wang et al., 2025b) and the mirage of test-time scaling (Ghosal et al., 2026), which further motivate principled control over reasoning effort.

Despite their progress, existing ALRMs share a fundamental limitation: reasoning effort is regulated through **discrete specified modes**, supported by coarse supervision signals such as external estimators, predefined data categories, or heuristic length constraints. Such discrete control overlooks the internal logical sufficiency of reasoning trajectories and lacks the flexibility to finely calibrate reasoning depth in a problem-specific manner.

In contrast, our work proposes *continuous adaptive reasoning* grounded in the concept of *Minimal Sufficient CoT* (MSC). By introducing a principled sufficiency criterion, we enable fine-grained assessment of whether a reasoning prefix is adequate to support a confident answer. Unlike discrete modes or fixed truncation rules, our sufficiency-aware training empowers the model to autonomously calibrate its reasoning effort along a continuous spectrum.

## 3. Methodology

### 3.1. Problem Formulation

**Notation.** Consider a dataset $\mathcal{D}$ of question-answer pairs $(x, y^*)$, where $x$ denotes an input question and $y^*$ the ground-truth answer. Given $x$, a reasoning model $\pi_\theta$ generates a CoT trajectory $z = (z_1, z_2, \ldots, z_{L_z})$, with $L_z$ sentences and a total of $\|z\|$ tokens. Conditioned on $x$ and $z$, the model generates the final answer: $y \sim \pi_\theta(\cdot \mid x, z)$.

### 3.2. MSC: Minimal Sufficient CoT

Figure 2 provides an intuitive illustration of MSC.

**Reasoning Sufficiency.** To quantify how well a reasoning trajectory supports the ground-truth, we define the *reasoning sufficiency*:

$$\mathcal{S}_\theta(z \mid x, y^*) := \left( \prod_{i=1}^{\|y^*\|} \pi_\theta(y_i^* \mid x, z, y_{<i}^*) \right)^{1/\|y^*\|} \quad (1)$$

The most natural signal is the joint probability $\prod_i \pi_\theta(y_i^* \mid x, z, y_{<i}^*)$. However, it decays exponentially with answer length, making it fragile for long sequences. To address this, we employ the geometric mean, which normalizes the joint probability into a per-token average. We empirically validate this choice in Appendix A.4.

**Sufficient CoT.** Then we can determine whether reasoning is adequate by introducing a confidence threshold $\delta \in [0, 1]$. A trajectory $z$ is termed $\delta$-*sufficient* if $\mathcal{S}_\theta(z|x, y^*) \geq \delta$.

**MSC Definition.** We further define the MSC as the shortest reasoning prefix satisfying sufficiency. We identify MSC

at the sentence level, as sentence boundaries naturally correspond to atomic reasoning steps, and avoid fragmentary truncation that may distort logical structure. We say the prefix $z_{<t^*}$ is a $\delta$-MSC if and only if:

$$\begin{cases} \mathcal{S}_\theta(z_{<t^*} \mid x, y^*) \geq \delta & \text{(Sufficiency)} \\ \mathcal{S}_\theta(z_{<t} \mid x, y^*) < \delta, \quad \forall t < t^* & \text{(Minimality)} \end{cases} \quad (2)$$

**Problem-Adaptive Threshold.** A fixed threshold $\delta$ applies the same confidence bar uniformly across all problems, regardless of their inherent difficulty. However, for simple problems, a high $\delta$ retains unnecessary reasoning, while for hard problems, a low $\delta$ may truncate critical reasoning steps prematurely. We therefore introduce a *problem-adaptive threshold*:

$$\delta(x) = \delta_0 + \alpha \cdot \mathcal{C}(x) \quad (3)$$

where $\delta_0$ is the base value, $\alpha$ controls sensitivity to complexity, and $\mathcal{C}(x) \in [0, 1]$ denotes problem complexity. This produces a more discriminative MSC distribution across difficulty levels, providing a stronger adaptive prior for subsequent training.

**Percentile-Based Complexity Estimation.** We estimate complexity as the percentile rank of its reasoning length in the dataset: reasoning length serves as a practical proxy for problem complexity, as empirically supported in Figure 1. Formally, given a dataset $\mathcal{D} = \{(x_i, y_i^*, z_i)\}_{i=1}^N$, we define:

$$\mathcal{C}(x_i) = \frac{1}{N} \sum_{j=1}^N \mathbb{1}[\|z_j\| \leq \|z_i\|] \quad (4)$$

This percentile-based measure is robust to outliers and yields values uniformly distributed in $[0, 1]$, ensuring stable threshold scaling across problems.

### 3.3. Stage I: MSC-Aligned Fine-Tuning

The first stage, termed MSC-Aligned Fine-Tuning (MFT), aligns the model to produce concise yet sufficient reasoning through SFT on a curated MSC dataset. This stage consists of two steps: (1) constructing MSC data from full CoT trajectories, and (2) fine-tuning the model to internalize adaptive reasoning patterns.

**MSC Data Construction.** From source dataset $\mathcal{D}_{\text{src}} = \{(x_i, y_i^*)\}_{i=1}^N$, a strong reasoning model $\mathcal{M}_{\text{LRM}}$ generates full CoT and answers: $(\hat{z}_i, \hat{y}_i) \sim \mathcal{M}_{\text{LRM}}(x_i)$. We then extract MSC from each trajectory via the following procedure:

▶ *(i) Compute adaptive thresholds.* With access to all trajectory lengths $\{\|\hat{z}_i\|\}_{i=1}^N$, we derive per-sample complexity $\mathcal{C}(x_i)$ and threshold $\delta(x_i)$ using Eq. 4 and Eq. 3.

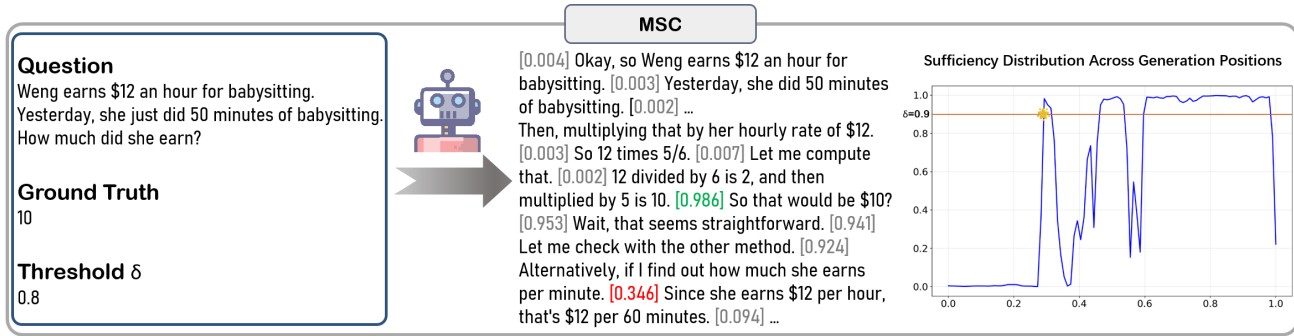

*Figure 2.* **Illustration of Minimal Sufficient CoT (MSC).** For a given question, sufficiency score (geometric mean over ground-truth answer tokens) is computed at each generation position. The MSC is the shortest prefix exceeding the adaptive threshold $\delta$. As shown, once the sufficiency threshold is reached, extended *waiting* or self-verification steps lead to a rapid decline in sufficiency, indicating that additional reasoning contributes little benefit and may even degrade confidence.

---

**Algorithm 1** MSC Dataset Construction

**Require:** Source dataset $\mathcal{D}_{\text{src}}$; models $\mathcal{M}_{\text{LRM}}, \mathcal{M}_{\text{refine}}, \pi_\theta$; hyperparameters $\delta_0, \alpha, L_{\min}$
1: **for** each $(x_i, y_i^*) \in \mathcal{D}_{\text{src}}$ **do**
2:     $(\hat{z}_i, \hat{y}_i) \sim \mathcal{M}_{\text{LRM}}(x_i)$
3: **end for**
4: $\mathcal{D}_{\text{full}} \leftarrow \{(x_i, y_i^*, \hat{z}_i, \hat{y}_i)\}_{i=1}^N$
5: **for** each $i \in [1, N]$ **do**
6:     $\mathcal{C}(x_i) \leftarrow \frac{1}{N} \sum_{j=1}^N \mathbb{1}[\|\hat{z}_j\| \leq \|\hat{z}_i\|]$
7:     $\delta(x_i) \leftarrow \delta_0 + \alpha \cdot \mathcal{C}(x_i)$
8:     $t^* \leftarrow \arg\min_{t \in [0, L_{\hat{z}_i}]} \mathcal{S}_\theta(\hat{z}_{i,<t} \mid x_i, y_i^*) \geq \delta(x_i)$
9:     **if** no such $t$ exists **then**
10:         $t_i^* \leftarrow \arg\max_{t \in [0, L_{\hat{z}_i}]} \mathcal{S}_\theta(\hat{z}_{i,<t} \mid x_i, y_i^*)$
11:     **end if**
12:     $z_i^{\text{raw}} \leftarrow \hat{z}_{i,<t_i^*}$
13:     **if** $L_{z_i^{\text{raw}}} \leq L_{\min}$ **then**
14:         $z_i^{\text{MSC}} \leftarrow \varnothing$
15:     **else**
16:         $z_i^{\text{MSC}} \leftarrow \mathcal{M}_{\text{refine}}(x_i, z_i^{\text{raw}}, \hat{y}_i)$
17:     **end if**
18: **end for**
19: **return** $\mathcal{D}_{\text{MSC}} = \{(x_i, z_i^{\text{MSC}}, \hat{y}_i)\}_{i=1}^N$

---

▶ *(ii) Identify raw MSC prefixes.* For each sample, we scan sentence-level prefixes to find the minimal sufficient one:

$$t_i^* = \underset{t \in [0, L_{\hat{z}_i}]}{\arg\min} \mathcal{S}_\theta(\hat{z}_{i,<t} \mid x_i, y_i^*) \geq \delta(x_i) \quad (5)$$

If no prefix reaches $\delta(x_i)$, we select the most sufficient one:

$$t_i^* = \underset{t \in [0, L_{\hat{z}_i}]}{\arg\max} \mathcal{S}_\theta(\hat{z}_{i,<t} \mid x_i, y_i^*). \quad (6)$$

This yields a raw candidate: $z_i^{\text{raw}} = \hat{z}_{i,<t_i^*}$.

To avoid trivial fragments, we set $z_i^{\text{raw}}$ with empty string if $\|z_i^{\text{raw}}\| \leq L_{\min}$, indicating that the question requires no explicit reasoning.

▶ *(iii) Refine MSC for coherence.* Raw truncation may leave logical gaps. We use $\mathcal{M}_{\text{refine}}$ to polish each nonempty

MSC with the following objectives: (1) naturally derive the answer, (2) eliminate redundancy, and (3) preserve stylistic consistency. This produces the final refined $z_i^{\text{MSC}}$.

The final dataset is formatted as:

$$\mathcal{D}_{\text{MSC}} = \left\{\left(x_i, \texttt{<think>}\, z_i^{\text{MSC}}\, \texttt{</think>}\, \hat{y}_i\right)\right\}_{i=1}^N \quad (7)$$

where $z_i^{\text{MSC}}$ can be empty for questions requiring no reasoning. The complete procedure is detailed in Algorithm 1.

**Supervised Fine-Tuning.** We fine-tune the base model by minimizing the negative log-likelihood over $\mathcal{D}_{\text{MSC}}$:

$$\mathcal{L}_{\text{MFT}}(\theta) = -\mathbb{E}_{(x_i, z_i^{\text{MSC}}, \hat{y}_i) \sim \mathcal{D}_{\text{MSC}}}$$
$$\left[ \log \pi_\theta(z_i^{\text{MSC}} \mid x_i) + \log \pi_\theta(\hat{y}_i \mid x_i, z_i^{\text{MSC}}) \right] \quad (8)$$

### 3.4. Stage II: Sufficiency-Aware Policy Optimization

In the second stage, named Sufficiency-Aware Policy Optimization (SAPO), we train the model to allocate reasoning steps during inference through RL with a dynamic complexity pool and sufficiency-aware rewards. We build upon Group Relative Policy Optimization (GRPO) (Shao et al., 2024), which samples multiple trajectories per question to enable robust group-wise advantage estimation.

**Dynamic Complexity Pool.** A critical challenge in integrating MSC into online RL is that the reasoning length distribution shifts as the policy evolves. The offline complexity estimates from MFT stage become obsolete. Recomputing them over the entire dataset after each gradient step is computationally prohibitive.

Instead, we maintain an online *dynamic complexity pool* $\mathcal{P} = \{\|z_i^{\text{avg}}\|\}_{i=1}^N$ that tracks the evolving reasoning length for each question $x_i$. The pool is initialized from $\pi_{\text{MFT}}$

on the RL training data, i.e., $\|z_i^{\text{avg}}\| \leftarrow \mathbb{E}_{z \sim \pi_{\text{MFT}}(\cdot|x_i)}[\|z\|]$. For each training batch, we update the pool via exponential moving average (EMA):

$$\|z_i^{\text{avg}}\| \leftarrow (1 - \eta) \cdot \|z_i^{\text{avg}}\| + \eta \cdot \frac{1}{K} \sum_{k=1}^{K} \|z_i^{(k)}\|, \quad (9)$$

where $\eta \in [0, 1]$ controls the update rate, and $\{z_i^{(k)}\}_{k=1}^K$ are the $K$ rollout trajectories for $x_i$ in the current batch.

From $\mathcal{P}$, we recompute complexity scores $\mathcal{C}(x_i)$ and thresholds $\delta(x_i)$ via Eq. 4 and Eq. 3. This mechanism ensures sufficiency targets aligned with the policy's current behavior, providing stable reward signals at negligible extra cost.

**Sufficiency-Aware Reward Shaping.** The total reward:

$$\mathcal{R}(z, y \mid x, y^*) = \mathcal{R}_{\text{cor}}(y) + \mathcal{R}_{\text{format}}(z, y) + \beta \cdot \mathcal{R}_{\text{suff}}(z \mid x, y^*) \tag{10}$$

where $\mathcal{R}_{\text{cor}}$ rewards correct answers, and $\mathcal{R}_{\text{format}}$ ensures proper use of `<think>...</think>` delimiters.

The sufficiency reward $R_{\text{suff}}$ uses the current adaptive threshold $\delta(x)$ from the dynamic pool. For each trajectory $z$, we identify the earliest sufficient prefix $z_{<t_i^*}$ using Eq. 5. If no prefix satisfies the threshold, we set $t^* = \infty$. The reward penalizes both over-thinking and under-thinking:

$$\mathcal{R}_{\text{suff}}(x, z, y) = \underbrace{-\lambda_{over} \cdot \mathbb{1}[L_z > t^* + \epsilon]}_{\text{over-thinking}}$$
$$- \underbrace{\mathbb{1}[y \neq y^*] \cdot \lambda_{under} \cdot \mathbb{1}[L_z < t^*]}_{\text{under-thinking}} \tag{11}$$

The tolerance $\epsilon$ allows minor deviations beyond sufficiency, and the under-thinking penalty applies only to incorrect generations.

## 4. Experiments

### 4.1. Experimental Settings

**Training datasets.** We train SuCo on reasoning datasets spanning mathematics, code, and science. The data are drawn from five sources: Llama-Nemotron Post-Training Dataset (Bercovich et al., 2025), Mixture-of-Thoughts (Hugging Face, 2025), OpenR1-Math-220k (Lozhkov et al., 2025), OpenCodeReasoning (Ahmad et al., 2025), and s1K-1.1 (Muennighoff et al., 2025). These datasets contain reasoning chains distilled from state-of-the-art LRMs. After filtering and deduplication, we construct the corresponding MSC for each sample following Algorithm 1. We further remove low-quality MSC samples using LLM-based quality assessment. Both MSC refinement and quality assessment are performed with Qwen3-Next-80B-A3B-Instruct (Qwen Team, 2025). This process yields 270,011 high-quality training samples. Detailed construction procedures and statistics

are provided in Appendix B. Figure 3 compares the token length distributions of full CoT and MSC across the training corpus. We use the full MSC dataset for Stage I, and sample a subset of the data for RL in Stage II.

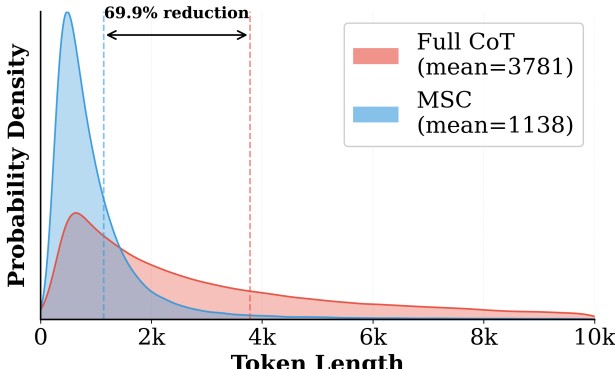

*Figure 3.* Token length distribution comparison between full CoT and MSC across training datasets.

**Implementation details.** All trainings are performed on $8 \times$ NVIDIA H100 80GB GPUs. **MFT Stage.** We set the base threshold $\delta_0 = 0.5$ and the sensitivity coefficient $\alpha = 0.4$, resulting in problem-adaptive thresholds $\delta(x) \in [0.5, 0.9]$. The minimum reasoning length is fixed to $L_{\min} = 5$ sentences to filter trivial fragments. We train for 3 epochs with a learning rate of $1 \times 10^{-4}$. **SAPO Stage.** The dynamic complexity pool is initialized using predictions from the MFT model and updated during training with an EMA rate $\eta = 0.1$. For each training instance, we sample $K = 8$ rollout trajectories. The sufficiency reward weight is set to $\beta = 1.0$, with over- and under-thinking penalties $\lambda_{\text{over}} = \lambda_{\text{under}} = 0.5$ and a tolerance margin $\epsilon = 2$ sentences. We train using Group Relative Policy Optimization (GRPO) (Shao et al., 2024) with learning rate $1 \times 10^{-6}$, a batch size of 128 and a micro batch size of 8.

**Benchmarks and Metrics.** We conduct comprehensive evaluations across mathematics, code, and science domains, covering a broad range of problem difficulties. For **mathematics**, we evaluate on GSM8K (Cobbe et al., 2021), MATH-500 (Lightman et al., 2024), AMC 2023, and AIME 2025. Due to the limited size of AMC 2023 (40 problems) and AIME 2025 (30 problems), each evaluation is repeated 10 times and results are averaged to reduce variance and improve statistical reliability. For **code**, we use MBPP (Austin et al., 2021), and LiveCodeBench v6 (Jain et al., 2025). For **science**, we test on MMLU-STEM (Hendrycks et al., 2021a) and GPQA-Diamond (Rein et al., 2024). Across all benchmarks, we report both accuracy and response length.

*Table 1.* Main results on mathematics (GSM8K, MATH-500, AMC23, AIME25), code (MBPP, LiveCodeBench-V6), and science (MMLU-STEM, GPQA-Diamond) benchmarks. Best results in each section are **bolded**, second best are underlined.

| Methods | Math | | | | Code | | Science | | Avg. |
|---|---|---|---|---|---|---|---|---|---|
| | GSM8K | MATH500 | AMC23 | AIME25 | MBPP | Live-V6 | MMLU-S | GPQA-D | |
| **(I) Reasoning Correctness Evaluation: Response Accuracy (%) ↑** | | | | | | | | | |
| *Qwen2.5-1.5B* | | | | | | | | | |
| Math-Base | 40.1 | 22.6 | 23.9 | 3.3 | 4.0 | 0.6 | 14.5 | 4.0 | 14.1 |
| Math-Instruct | 79.0 | 72.4 | 43.5 | 6.7 | 6.1 | 2.3 | 30.2 | 22.2 | 32.8 |
| DeepSeek-R1-Distill | 80.3 | 80.6 | 56.5 | 26.7 | 41.0 | 17.1 | 33.8 | 25.3 | 45.2 |
| AdaCoT | 82.7 | 83.2 | 62.5 | 27.3 | 44.3 | 17.7 | 33.9 | 26.3 | 47.2 |
| AdaptThink | 83.2 | 83.8 | 65.8 | 28.3 | 42.1 | 18.3 | 34.4 | 26.8 | 47.8 |
| S-GRPO | 83.4 | 84.2 | 69.5 | 31.0 | 42.9 | 19.4 | 34.8 | 28.3 | 49.2 |
| LHRMs | 85.7 | 84.4 | 70.0 | **34.0** | 43.1 | 20.6 | 35.7 | 30.3 | 50.5 |
| **SuCo (Ours)** | **87.7** | **86.8** | **73.8** | 33.7 | **48.5** | **22.3** | **38.6** | **33.3** | **53.1** |
| *Qwen2.5-7B* | | | | | | | | | |
| Math-Base | 61.8 | 54.2 | 36.2 | 7.0 | 13.1 | 1.1 | 26.4 | 13.1 | 26.6 |
| Math-Instruct | 87.0 | 72.4 | 53.5 | 12.3 | 26.4 | 8.0 | 51.3 | 32.3 | 42.9 |
| DeepSeek-R1-Distill | 89.3 | 89.0 | 75.5 | 49.7 | 57.6 | 31.4 | 67.6 | 45.5 | 63.2 |
| AdaCoT | 91.4 | 91.8 | 81.5 | 55.0 | 61.7 | 32.0 | 69.0 | 47.0 | 66.2 |
| AdaptThink | 92.9 | 92.8 | 82.0 | 54.3 | 62.0 | 32.6 | 68.8 | 47.0 | 66.6 |
| S-GRPO | 92.8 | 92.2 | **90.5** | 58.3 | 62.3 | 35.4 | 72.2 | 51.5 | 69.4 |
| LHRMs | 92.4 | 93.0 | 87.3 | 57.7 | 62.4 | 35.4 | 71.4 | 49.5 | 68.6 |
| **SuCo (Ours)** | **93.9** | **93.6** | 90.3 | **61.7** | **65.7** | **38.9** | **75.8** | **56.6** | **72.1** |
| **(II) Reasoning Efficiency Evaluation: Response Length (Tokens) ↓** | | | | | | | | | |
| *Qwen2.5-1.5B* | | | | | | | | | |
| DeepSeek-R1-Distill | 501 | 4,260 | 6,768 | 11,239 | 3,511 | 11,073 | 1,956 | 6,582 | 5,736 |
| AdaCoT | 443 | 1,479 | 2,936 | 6,271 | 1,720 | 6,455 | 1,029 | 3,279 | 2,952 |
| AdaptThink | 337 | 1,564 | 2,740 | 6,513 | 1,422 | 6,689 | 995 | 2,914 | 2,897 |
| S-GRPO | 297 | 1,377 | 3,081 | 6,640 | 1,481 | 5,293 | 812 | 2,828 | 2,726 |
| LHRMs | **242** | 1,252 | 2,477 | **3,257** | 1,158 | 4,716 | 955 | 2,381 | 2,055 |
| **SuCo (Ours)** | 304 | **538** | **1,687** | 3,484 | **930** | **2,629** | **745** | **1,550** | **1,483** |
| *Qwen2.5-7B* | | | | | | | | | |
| DeepSeek-R1-Distill | 465 | 3,126 | 5,466 | 10,833 | 3,182 | 10,407 | 1,572 | 6,858 | 5,239 |
| AdaCoT | 422 | 1,387 | 3,115 | 7,648 | 1,423 | 8,153 | 1,214 | 3,993 | 3,419 |
| AdaptThink | 247 | 1,424 | 2,834 | 7,842 | 1,242 | 8,911 | 1,023 | 3,677 | 3,400 |
| S-GRPO | 267 | 988 | 2,247 | 5,259 | 1,356 | 6,534 | 717 | 2,453 | 2,478 |
| LHRMs | 274 | 658 | 1,525 | 4,217 | 1,487 | 4,294 | 628 | 2,042 | 1,891 |
| **SuCo (Ours)** | **243** | **429** | **935** | **2,679** | **1,149** | **2,809** | **505** | **1,389** | **1,267** |

**Baselines.** We implement SuCo on Qwen2.5-Math-1.5B/7B-Base (Yang et al., 2024) and compare against the following baselines at matched model scales. **Standard Models.** We evaluate Qwen2.5-Math-Base, Qwen2.5-Math-Instruct, along with DeepSeek-R1-Distill-Qwen (Guo et al., 2025) as the full CoT reasoning baseline. **Adaptive Large Reasoning Models (ALRMs).** We compare with four representative ALRMs: (1) AdaCoT (Lou et al., 2025) employs an external complexity assessor and PPO with Pareto optimization. (2) AdaptThink (Zhang et al., 2025b) uses constrained RL for binary mode selection. (3) S-GRPO (Dai et al., 2025) samples multiple early-exit positions with decaying rewards during RL training. (4) LHRMs (Jiang et al., 2025) performs hybrid fine-tuning on categorized data followed by group policy optimization. For fair comparison, AdaCoT and LHRMs are initialized from Qwen2.5-Math-Base and trained on the same source data as SuCo, while AdaptThink and S-GRPO follow their original implementa-

tions using DeepSeek-R1-Distill-Qwen as the base model.

## 4.2. Main Results

**Reasoning Correctness Evaluation.** As shown in Table 1 (I), across all model scales and domains, SuCo consistently achieves the highest or near-highest accuracy. At the 1.5B scale, SuCo attains an accuracy of 53.1%, achieving a relative improvement of 5.1% over the strongest adaptive baseline LHRMs and 17.5% over DeepSeek-R1-Distill-Qwen. At the 7B scale, SuCo further improves to 72.1% accuracy, exceeding LHRMs by 5.1% and DeepSeek-R1-Distill-Qwen by 14.1%. Notably, SuCo exhibits particularly strong gains on challenging benchmarks. For example, on AIME25, SuCo attains 33.7% accuracy at 1.5B scale and 61.7% at 7B scale, corresponding to relative improvements of 26.2% and 24.1% over DeepSeek-R1-Distill-Qwen, respectively.

*Table 2.* Ablation results of MFT components. The results evaluate the effectiveness of MFT against the base model and full CoT training. We further analyze the sensitivity of sufficiency thresholds, complexity estimation strategies, and the impact of MSC refinement.

| Component | Method | Math | | Code | | Science | | Avg. | |
|---|---|---|---|---|---|---|---|---|---|
| | | Acc ↑ | Tokens ↓ | Acc ↑ | Tokens ↓ | Acc ↑ | Tokens ↓ | Acc ↑ | Tokens ↓ |
| Overall | Base | 22.5 | - | 2.3 | - | 9.3 | - | 11.4 | - |
| | Full | 59.1 | 5,380 | 28.5 | 6,345 | 27.7 | 3,522 | 38.4 | 5,082 |
| | **MFT** | **69.1** | **1,359** | **33.2** | **1,706** | **34.8** | **966** | **45.7** | **1,344** |
| Threshold | $\delta = 0.9$ | 66.8 | 1,545 | 30.6 | 2,128 | 32.0 | 1,308 | 43.1 | 1,660 |
| | $\delta = 0.8$ | 66.1 | 1,420 | 30.1 | 1,853 | 31.0 | 1,072 | 42.4 | 1,448 |
| | $\delta = 0.7$ | 67.2 | 1,246 | 31.0 | 1,769 | 33.8 | 1,024 | 44.0 | 1,346 |
| | $\delta = 0.6$ | 60.9 | 1,072 | 26.6 | 1,645 | 27.5 | 910 | 38.3 | 1,209 |
| | $\delta = 0.5$ | 61.9 | 889 | 25.7 | 1,305 | 25.8 | 771 | 37.8 | 988 |
| Complexity | Min–Max | 61.7 | 1,130 | 26.1 | 1,506 | 27.9 | 843 | 38.6 | 1,160 |
| | Log-Scaled | 68.4 | 1,506 | 31.5 | 1,642 | 33.5 | 1,049 | 44.5 | 1,399 |
| Refinement | w/o refine | 65.9 | 2,220 | 31.8 | 2,741 | 32.7 | 1,607 | 43.5 | 2,189 |

**Reasoning Efficiency Evaluation.** Table 1 (II) reports reasoning efficiency measured by average response length. In addition to attaining higher accuracy, SuCo significantly reduces token consumption compared to DeepSeek-R1-Distill-Qwen. Across all benchmarks, SuCo reduces average response tokens by 74.1% at the 1.5B scale and by 75.8% at the 7B scale, yielding substantial inference cost savings. On AIME25 at 7B scale, SuCo achieves 24.1% higher accuracy while using 75.3% fewer tokens. SuCo also outperforms other adaptive reasoning methods, confirming that sufficiency-aware training eliminates redundant reasoning without sacrificing decision quality.

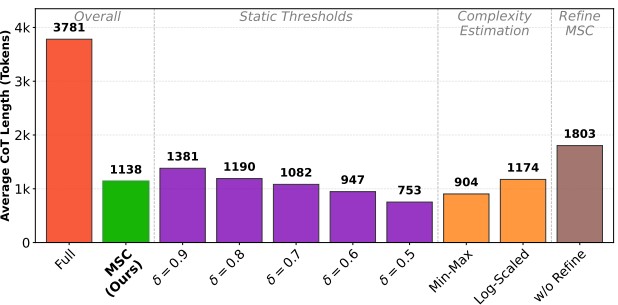

*Figure 4.* Distribution of reasoning lengths in training data constructed by different MSC variants.

### 4.3. Ablation Study

We analyze the contribution of each component in SuCo on Qwen2.5-Math-1.5B. Additional ablation studies on hyperparameters ($L_{\min}$, $\epsilon$, $\eta$) are provided in Appendix A.

**MFT Ablations.** Results are summarized in Table 2, with training CoT length distributions illustrated in Figure 4.

▶ *Overall Effectiveness.* While full CoT training improves the base model from 11.4% to 38.4% accuracy, it generates verbose reasoning. In contrast, MFT achieves higher accuracy at 45.7% while consuming only 26.4% of full CoT's reasoning overhead. This confirms that MSC is not merely compressed reasoning but a more effective form that filters noise and streamlines logical flow, enabling better performance with significantly reduced computational cost.

▶ *Problem-Adaptive Threshold.* Static thresholds $\delta \in [0.5, 0.9]$ exhibit clear accuracy-efficiency trade-offs. High thresholds retain excessive reasoning, while low thresholds sacrifice critical reasoning steps, leading to noticeable performance degradation. Among static settings, $\delta = 0.7$ achieves the best balance. Nevertheless, problem-adaptive thresholds naturally align with problem demands, surpass-

ing all static configurations with comparable token usage.

▶ *Percentile-Based Complexity Estimation.* We compare against two alternatives: Min-Max estimation $\mathcal{C}(x_i) = \frac{(\|z_i\| - \min_j \|z_j\|)}{(\max_j \|z_j\| - \min_j \|z_j\|)}$ and Log-Scaled normalization $\mathcal{C}(x_i) = \frac{\log(1 + \|z_i\|) - \log(1 + \min_j \|z_j\|)}{\log(1 + \max_j \|z_j\|) - \log(1 + \min_j \|z_j\|)}$. Min-Max estimation is highly sensitive to outliers, a single extremely long trajectory compresses all other samples into a narrow range, resulting in poor complexity discrimination. Log-Scaled normalization partially alleviates this issue but still results in skewed scaling. In contrast, percentile-based method produces a uniform complexity distribution, ensuring stable threshold scaling across diverse problems.

▶ *MSC Refinement.* Without refinement, directly truncated CoT trajectories often results in abrupt or incomplete logical transitions. The refinement process bridges these logical gaps while simultaneously eliminating redundancy, producing more coherent and concise reasoning chains. Consequently, refinement reduces reasoning length by 38.6% and boosts accuracy by 5.1%. Prompts along with concrete examples are provided in Appendix C.

*Table 3.* Ablation study of SAPO components. Dynamic Complexity Pool (DCP) and Sufficiency-Aware Reward Shaping($R_{suff}$).

| Method | Accuracy (%) ↑ | Response Tokens ↓ |
|---|---|---|
| MFT | 51.5 | 1,347 |
| **SAPO** | 53.1 | 1,483 |
| w/o DCP | 52.9 | 1,642 |
| w/o $R_{suff}$ | 52.7 | 2,053 |

**SAPO Ablations.** We summarize ablation results in Table 3 and visualize the per-benchmark accuracy-efficiency trade-offs in Figure 5.

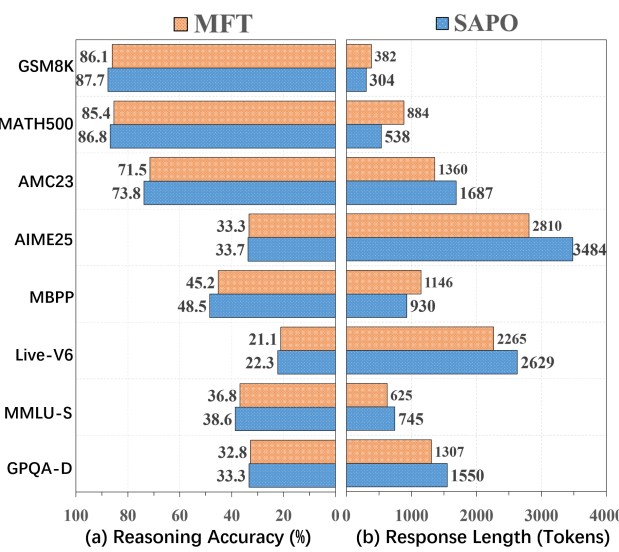

*Figure 5.* Per-benchmark accuracy and response length comparison. SAPO adaptively reduces reasoning on easier benchmarks while allocating more resources to challenging ones.

▶ *Overall Effectiveness.* Although SAPO slightly increases the response length, it improves accuracy across all benchmarks. Crucially, this response increase does not reflect redundant reasoning. As shown in Figure 5, on simple benchmarks where MFT already achieves high accuracy, SAPO successfully reduces reasoning. Conversely, on challenging benchmarks, SAPO intelligently allocates additional reasoning budget. This behavior indicates that SAPO learns to calibrate reasoning effort based on problem demands.

▶ *Dynamic Complexity Pool.* In this ablation(w/o DCP), the complexity pool is initialized using MFT predictions but remains fixed during RL training. Without online EMA updates, the estimated complexity gradually drifts away from the evolving policy. This misalignment results in stale thresholds that fail to provide accurate sufficiency targets.

▶ *Sufficiency-Aware Reward Shaping.* When the sufficiency reward is removed (w/o $R_{suff}$), SAPO degenerates to vanilla GRPO that optimizes only correctness and for-

mat, collapsing to verbose, full-CoT-style reasoning patterns. The sufficiency reward provides fine-grained feedback on both over-thinking and under-thinking, encouraging concise yet reliable reasoning.

## 4.4. Analysis

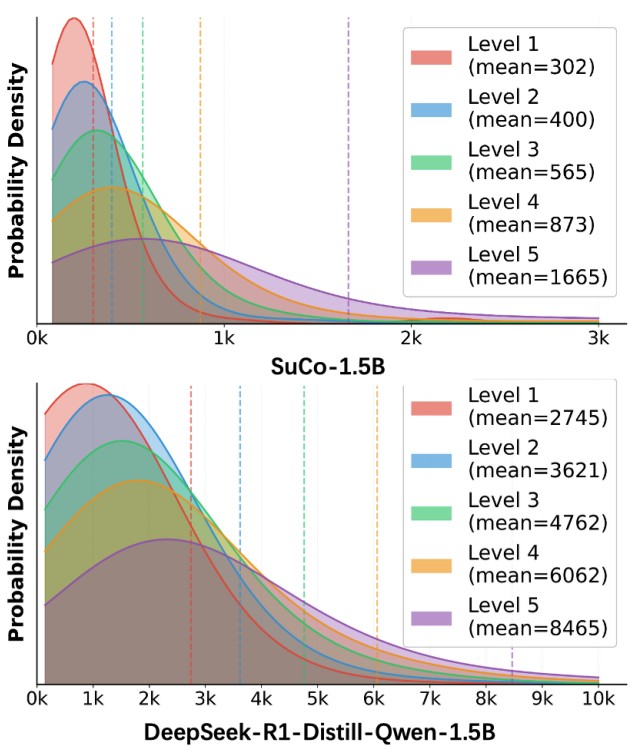

*Figure 6.* Response length distribution across MATH difficulty levels for SuCo-1.5B (top) and base LRM DeepSeek-R1-Distill-1.5B (bottom). SuCo continuously adapts reasoning effort to problem complexity with significantly higher efficiency.

**Difficulty-conditioned reasoning length.** We compare response length distributions across MATH (Hendrycks et al., 2021b) difficulty levels between SuCo-1.5B and DeepSeek-R1-Distill-1.5B. As shown in Figure 6, both models shift rightward as difficulty increases, but SuCo exhibits a much higher difficulty-sensitivity ratio: the Level 5/Level 1 mean token ratio is $\approx 5.5\times$ for SuCo versus $\approx 3.1\times$ for the base LRM, indicating more discriminative resource allocation.

Moreover, SuCo operates in a fundamentally more efficient regime. On Level 1 problems, it uses 89% fewer tokens than the base LRM while maintaining accuracy. The base LRM's length variation reflects an inability to truncate unnecessary reasoning even for trivial queries, whereas SuCo's variation reflects genuine difficulty-conditioned allocation learned through sufficiency-aware training.

**Out-of-Domain Generalization.** To assess whether SuCo's adaptive reasoning capability generalizes beyond

the training domains, we conduct out-of-domain (OOD) evaluations on StrategyQA (Geva et al., 2021), CommonsenseQA (Talmor et al., 2019), and AlpacaEval 2.0 (Li et al., 2023). These tasks differ from the training distribution.

*Table 4.* Out-of-domain generalization results. SuCo demonstrates strong transfer of adaptive reasoning to unseen task types.

| Method | StrategyQA ACC / Tok | CSQA ACC / Tok | AlpacaEval LC-WR / Tok |
|---|---|---|---|
| DeepSeek-R1 -Distill | 53.3 / 483 | 45.0 / 743 | 1.05 / 596 |
| Full CoT SFT | 22.6 / 742 | 19.4 / 1,061 | 0.3 / 743 |
| MFT | 28.0 / 213 | 26.6 / 342 | 0.67 / 314 |
| **SuCo** | **55.7** / 442 | **49.3** / 369 | **2.4** / 288 |

As shown in Table 4, SuCo substantially outperforms all baselines on OOD tasks. Notably, while MFT alone overfits to training domain patterns and degrades on OOD tasks, the SAPO stage enables SuCo to learn a generalizable policy for calibrating reasoning effort.

**Cross-Model Robustness of MSC.** To verify that MSC boundaries are robust across model families, we construct MSC data using different calibrator models (Qwen3-4B, Qwen3-14B, DeepSeek-R1-Distill-Qwen-7B) and train different target models. As shown in Table 5, MSC supervision from all calibrators consistently outperforms full-CoT training across target models, confirming that the constructed datasets transfer well across model families.

*Table 5.* Cross-model robustness. Different calibrator models produce MSC data that consistently improves over Full CoT SFT across target model families. Format: Accuracy (Tokens).

| Calibrator | Qwen2.5-1.5B | Llama-3.2-3B |
|---|---|---|
| Full CoT SFT | 37.4 (5,177) | 38.1 (5,084) |
| Qwen3-4B | 44.7 (1,394) | 44.2 (1,524) |
| Qwen3-14B | 44.2 (1,521) | 43.2 (1,821) |
| DS-R1-Distill-7B | 44.5 (1,491) | 43.7 (1,691) |

**Empty CoT Analysis.** SuCo learns to skip explicit reasoning when problems are trivial, directly outputting answers without explicit reasoning. Figure 7(a) reveals that empty CoT rates decrease monotonically with increasing difficulty, indicating that the model increasingly engages in explicit reasoning for harder problems. The 7B model consistently exhibits higher empty rates than 1.5B across all levels, reflecting its stronger capabilities that reduce reliance on intermediate reasoning steps.

Across domains (Figure 7(b)), empty CoT rates remain relatively stable at 30–37%, suggesting that the decision to omit explicit reasoning is largely task-agnostic. Math problems show a slightly higher proportion of empty responses, likely due to formula-based questions requiring minimal explicit

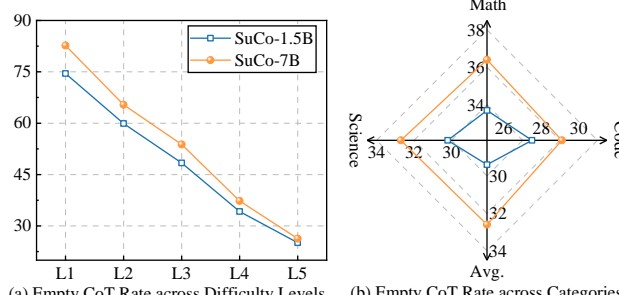

(a) Empty CoT Rate across Difficulty Levels  (b) Empty CoT Rate across Categories

*Figure 7.* Empty CoT analysis of SuCo-1.5B and SuCo-7B across problem types and difficulties. Higher model capacity (7B vs. 1.5B) leads to increased empty CoT rates, while harder problems trigger more explicit reasoning.

derivation. Despite a substantial fraction of empty CoT outputs, SuCo maintains strong overall accuracy (Table 1), suggesting that explicit reasoning is not always necessary, and that selectively omitting CoT can preserve or even improve efficiency without sacrificing accuracy.

## 5. Conclusion

In this work, we formalize *Minimal Sufficient CoT* (MSC) as the shortest reasoning prefix adequate for correct answers, revealing that models can perform better with less reasoning. Building on this insight, we propose *Sufficiency-guided Continuous Adaptive Reasoning* (SuCo), a two-stage framework enabling continuous and autonomous reasoning adaptation. Through *MSC-Aligned Fine-Tuning* (MFT) and *Sufficiency-Aware Policy Optimization* (SAPO), SuCo learns to calibrate its reasoning effort according to problem demands without relying on discrete modes or external controllers. Extensive experiments across mathematics, code, and science benchmarks demonstrate that SuCo consistently achieves higher accuracy with significantly fewer reasoning tokens.

**Limitations.** We acknowledge several limitations. First, MSC construction relies on ground-truth answers to compute sufficiency scores, which limits direct application to open-ended generation tasks. However, once trained, the model internalizes adaptive reasoning as a general capability. Second, the MFT stage depends on data distilled from strong LRMs. While removing the 80B refinement model still yields results superior to all baselines, reducing this dependency remains desirable.

**Future Work.** Extending sufficiency estimation to open-ended settings is a promising avenue. Additionally, agentic tasks present a compelling application scenario, where overthinking incurs redundant API costs and under-thinking leads to task failure. Extending SuCo to such settings is a promising direction.

## Impact Statement

This work aims to advance the field of machine learning by proposing a more efficient and adaptive training framework for reasoning models. Our method focuses on technical efficiency improvements and does not alter the fundamental capabilities or safety properties of underlying models. We do not foresee any ethical concerns or societal consequences beyond those commonly associated with research on large language models.

## Acknowledgements

This work was supported in part by National Key R&D Program of China (2024YFE0215300), National Natural Science Foundation of China (62476070), Shenzhen Science and Technology Program (JCYJ20241202123503005, GXWD20231128103232001, ZDSYS20230626091203008, KQTD20240729102154066), Department of Science and Technology of Guangdong (2024A1515011540) and National Key R&D Program of China (SQ2024YFE0200592).

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

# A. Additional Ablation Studies

## A.1. Minimum Reasoning Threshold $L_{\min}$

During MSC construction, if the raw MSC prefix contains fewer than or equal to $L_{\min}$ sentences, we set it to an empty string, indicating the model should directly generate the answer without intermediate reasoning steps.

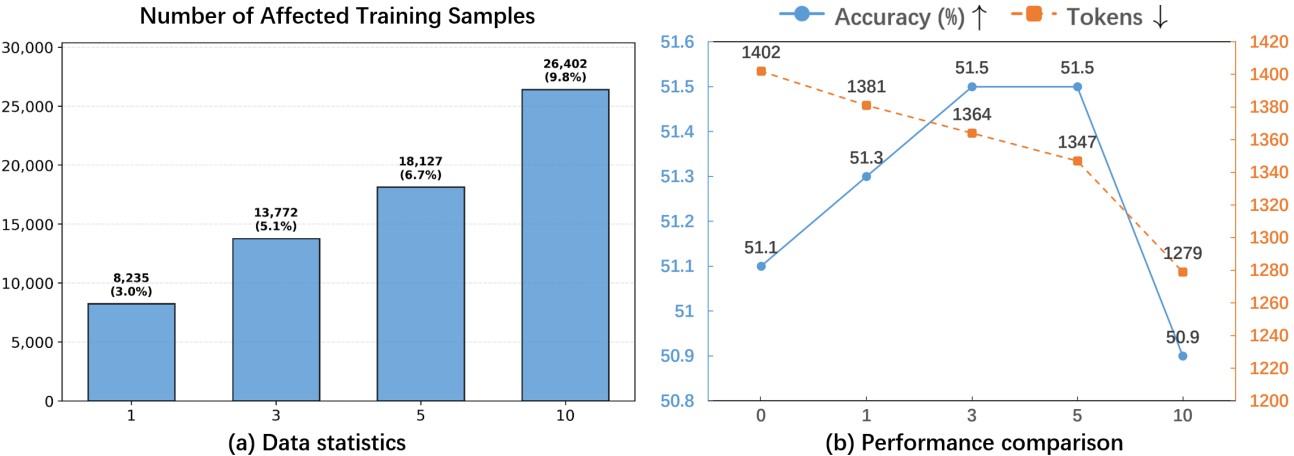

*Figure 8.* Effect of the minimum reasoning length $L_{\min}$.

Figure 8(a) shows the number of affected training samples at different thresholds. As $L_{\min}$ increases from 1 to 10, the proportion of non-thinking samples grows from 3.0% to 9.8%.

As shown in Figure 8(b), without the threshold ($L_{\min} = 0$), trivial CoT fragments introduce noise, resulting in 51.1% accuracy with 1,402 tokens. Overly aggressive filtering ($L_{\min} = 10$) suppresses necessary reasoning, degrading accuracy to 50.9%. $L_{\min} = 5$ achieves the optimal balance at 51.5% accuracy with 1,347 tokens, demonstrating that filtering very short CoT fragments (affecting 6.7% of samples) effectively removes noise while preserving meaningful reasoning signals.

## A.2. EMA Rate $\eta$.

We analyze the impact of EMA rate $\eta$ on the dynamic complexity pool update. As shown in Table 6, $\eta = 0.1$ achieves the best accuracy-efficiency balance at 53.1% accuracy with 1,483 tokens. Static pool ($\eta = 0$) retains more redundant reasoning (1,642 tokens) while achieving comparable accuracy (52.9%). Overly aggressive updates ($\eta \geq 0.5$) reduce token usage but degrade accuracy due to unstable threshold estimation. This validates that moderate EMA rates effectively balance tracking policy evolution with maintaining stable training signals.

*Table 6.* Effect of EMA rate $\eta$.

| EMA Rate ($\eta$) | Accuracy ↑ | Tokens ↓ |
|---|---|---|
| 0.0 (w/o DCP) | 52.9 | 1,642 |
| **0.1** | **53.1** | **1,483** |
| 0.3 | 52.9 | 1,442 |
| 0.5 | 52.6 | 1,427 |
| 1.0 (Full Update) | 52.1 | 1,369 |

## A.3. Over-thinking Tolerance $\epsilon$

The tolerance parameter $\epsilon$ in Eq. 11 controls the strictness of over-thinking penalties by allowing minor deviations beyond the minimal sufficient prefix.

As shown in Table 7, setting $\epsilon = 0$ applies strict penalties for any reasoning beyond the minimal sufficient prefix, resulting in overly aggressive truncation that reduces tokens to 1,391 but harms accuracy (52.4%). This strict constraint prevents the model from generating natural reasoning flow and exploring slightly longer but potentially more robust reasoning paths.

With moderate tolerance ($\epsilon = 2$), the model achieves the best accuracy at 53.1% while generating 1,483 tokens. This tolerance allows the

*Table 7.* Effect of over-thinking tolerance $\epsilon$ on SAPO performance. Results are averaged across all benchmarks on Qwen2.5-Math-1.5B.

| Tolerance ($\epsilon$) | Accuracy (%) ↑ | Tokens ↓ |
|---|---|---|
| 0 (Strict) | 52.4 | 1,391 |
| 1 | 52.8 | 1,456 |
| **2** | **53.1** | **1,483** |
| 3 | 53.0 | 1,527 |
| 5 | 52.7 | 1,658 |

model to extend reasoning by 1-2 sentences beyond the minimal
sufficient point when beneficial, accommodating natural variations in
reasoning style without sacrificing efficiency.

As $\epsilon$ increases further (3, 5), accuracy plateaus or slightly declines
while token usage grows substantially. At $\epsilon = 5$, the sufficiency constraint becomes too loose, allowing the model to generate verbose reasoning (1,658 tokens) that approaches the behavior without sufficiency-aware rewards. This demonstrates that $\epsilon = 2$ provides an appropriate balance: it avoids overly rigid constraints that harm reasoning quality while maintaining effective control over redundant thinking.

### A.4. Sufficiency Metric Ablation

We compare our geometric mean sufficiency formulation against alternative definitions to justify the design choice. All variants use the same MSC construction pipeline with Qwen2.5-Math-1.5B as the target model.

*Table 8.* Comparison of sufficiency metric formulations. Geometric mean provides the best balance between accuracy and efficiency.

| Sufficiency Metric | Train CoT Tokens | Accuracy (%) | Inference Tokens |
|---|---|---|---|
| Full CoT SFT (no truncation) | 3,781 | 38.4 | 5,082 |
| Joint Probability | 2,149 | 40.1 | 2,573 |
| Arithmetic Mean | 1,341 | 43.1 | 1,578 |
| **Geometric Mean (Ours)** | **1,138** | **45.7** | **1,344** |

**Joint Probability** ($\prod_i \pi_\theta(y_i^*|\cdot)$) decays exponentially with answer length, causing the threshold to be satisfied too late for short-answer problems and too early for long-answer problems. This results in inconsistent truncation quality.

**Arithmetic Mean** ($\frac{1}{\|y^*\|} \sum_i \pi_\theta(y_i^*|\cdot)$) is dominated by a few high-confidence tokens, making it less sensitive to tokens that genuinely require reasoning support.

**Geometric Mean** (Eq. 1) normalizes joint probability into per-token average log-probability, which is stable across varying answer lengths and equally sensitive to all answer tokens. It achieves the highest accuracy with the most aggressive token reduction, confirming its effectiveness as a sufficiency signal.

### A.5. Cross-Domain vs. Intra-Domain Percentile

In our default setting, complexity percentiles are computed globally across all training domains. However, different domains exhibit different baseline reasoning lengths. For instance, code problems typically require longer traces than math problems. This raises the question of whether a code problem might be assigned an artificially high complexity score simply because code traces are longer on average, rather than because the problem itself is harder. To investigate this, we compare the default cross-domain percentile with an intra-domain variant that computes percentiles separately within each domain (math, code, science).

*Table 9.* Cross-domain vs. intra-domain percentile estimation (Qwen2.5-Math-1.5B MFT).

| Method | Math | | Code | | Science | | Avg. | |
|---|---|---|---|---|---|---|---|---|
| | Acc | Tok | Acc | Tok | Acc | Tok | Acc | Tok |
| Full CoT SFT | 59.1 | 5,380 | 28.5 | 6,345 | 27.7 | 3,522 | 38.4 | 5,082 |
| Cross-domain | 69.1 | 1,359 | 33.2 | 1,706 | 34.8 | 966 | 45.7 | 1,344 |
| Intra-domain | 69.2 | 1,376 | 33.2 | 1,692 | 35.1 | 1,021 | 45.8 | 1,363 |

Intra-domain percentile yields nearly identical performance to the cross-domain setting, indicating that the global percentile preserves monotonicity within each domain and remains a robust measure of reasoning difficulty. This robustness arises because percentile ranks maintain relative ordering within domains, regardless of absolute length differences across domains.

# B. MSC Dataset Construction

## B.1. Dataset Statistics

Table 10 summarizes the statistics of the final dataset. Across all samples, the full CoT traces average 3,781 tokens, while MSC reduces this to 1,138 tokens. Notably, 109,882 samples (40.7%) yield empty MSCs, indicating that the model can solve these problems without explicit reasoning.

For Stage I, we train on all samples to learn minimal sufficient reasoning patterns. For Stage II, we sample 50,000 instances for RL to balance training efficiency and diversity.

*Table 10.* Training dataset statistics. We report the number of samples, average token counts for full CoT and MSC, and the number of samples with empty MSC (requiring no explicit reasoning).

| DOMAIN | SOURCE | SAMPLES | FULL CoT | RAW MSC | REFINE MSC | EMPTY |
|---|---|---|---|---|---|---|
| MATH | LLAMA-NEMOTRON | 39,377 | 2,924 | 1,238 | 1,067 | 10,683 |
| | MIXTURE-OF-THOUGHTS | 51,089 | 5,414 | 2,195 | 1,424 | 14,684 |
| | OPENR1-MATH-220K | 37,899 | 4,482 | 1,934 | 1,217 | 13,997 |
| | S1K-1.1 | 330 | 7,996 | 3,410 | 1,681 | 75 |
| CODE | LLAMA-NEMOTRON | 60,000 | 2,263 | 1,329 | 537 | 24,837 |
| | MIXTURE-OF-THOUGHTS | 12,671 | 11,106 | 2,764 | 2,379 | 2,914 |
| | OPENCODEREASONING | 15,704 | 6,076 | 2,491 | 1,599 | 3,424 |
| SCIENCE | MIXTURE-OF-THOUGHTS | 52,876 | 1,590 | 1,597 | 623 | 39,242 |
| | S1K-1.1 | 65 | 8,644 | 1,179 | 2,178 | 26 |
| TOTAL | | 270,011 | – | – | – | 109,882 |
| AVERAGE | | – | 3,781 | 1,803 | 1,138 | – |

## B.2. Data Sources

We curate our training data from the following five publicly available reasoning datasets, all containing CoT trajectories distilled from advanced LRMs:

- **Mixture-of-Thoughts** (Hugging Face, 2025): 350K samples (93K math, 83K code, 173K science) generated by DeepSeek-R1 with correctness filtering on final answers.

- **OpenR1-Math-220k** (Lozhkov et al., 2025): 220K math reasoning trajectories distilled from 800K DeepSeek-R1 generated solutions.

- **Llama-Nemotron Post-Training Dataset** (Bercovich et al., 2025): 3.9M samples covering math, code, science, chat, and safety. All samples include explicit reasoning trajectories produced by DeepSeek-R1 and refined using Nemotron-340B (Adler et al., 2024).

- **OpenCodeReasoning** (Ahmad et al., 2025): 735K Codeforces/LeetCode problems paired with CoT and executable Python solutions, including full test cases.

- **s1K-1.1** (Muennighoff et al., 2025): 1,000 carefully curated high-difficulty examples selected for difficulty, diversity, and quality, with accompanying budget-constrained inference technique.

## B.3. Data Preprocessing

We apply a rigorous preprocessing pipeline to ensure data quality:

**Filtering.** We remove samples with: (1) incorrect or missing answers, (2) incomplete reasoning traces, (3) overlap with our evaluation benchmarks, and (4) embedded non-textual elements (e.g., images, URLs), (5) non-English content.

**Deduplication.** We apply MinHash LSH (Lee et al., 2022) to remove near-duplicate samples.

**Cleaning.** Questions are normalized by removing source identifiers and numbering to reduce stylistic noise.

## B.4. MSC Construction

For each sample, we derive its MSC following Algorithm 1. Additionally, we employ an LLM-based evaluation to score each MSC along three dimensions: (1) correctness, (2) sufficiency and support for the final answer, (3) fluency and logical coherence. Low-quality MSC samples are filtered out. Both MSC refinement and quality assessment are performed with Qwen3-Next-80B-A3B-Instruct (Qwen Team, 2025).

# C. MSC Refinement

## C.1. Refinement Prompt

Figure 9 presents the complete prompt used for MSC refinement. The prompt guides the model to polish the raw MSC prefix along three dimensions: **Logical Completeness**, **Conciseness**, and **Stylistic Consistency**. The refinement process focuses on improving coherence and readability of the existing MSC without modifying its underlying reasoning content.

---

**MSC Refinement Prompt**

[System]
You are an expert Data Annotator specializing in mathematical reasoning and Chain-of-Thought (CoT) optimization.
Your task is to generate a complete, coherent, and concise CoT based on a truncated draft. You must follow these strict guidelines:

1. Seamless Completion:
   - You are provided with a `Draft CoT`. If the logic stops mid-stream, you must accept this prefix and continue the reasoning logic naturally until the final answer is derived.
   - The transition from the truncated part to your completion must be invisible and stylistically consistent.
   - If the `Draft CoT` already logically derives the `Target Answer`, DO NOT add more reasoning steps.

2. Target-Driven Reasoning:
   - You are provided with the `Target Answer`. You must ensure the reasoning logically and mathematically leads to this exact result.
   - Do NOT reveal that you know the Target Answer. The derivation must appear completely organic and deductive.

3. Optimization & Redundancy Removal:
   - No Loops: If the provided `Draft CoT` contains repetitive checks or stuck logic, you may subtly streamline it in your output, but keep the core "thinking" style.
   - Remove repetitive thinking, filler phrases, unnecessary "wait", "let me check", "reflect", or post-answer reflections from the output *unless* they are critical for a complex insight.
   - Stop at Success: Once the final result is derived and verified, STOP. Do not add unnecessary "wait," "let me think again," or excessive self-reflection after the solution is found.
   - If the `Draft CoT` contains "noise" at the very end (after the answer was reached), delete that noise.

4. Style Consistency:
   - Mimic the style, tone, and structure of the provided `Draft CoT`.
   - Maintain the structure of the original thought process but ensure it is efficient.

You will be given a Question, a Target Answer, and the Draft CoT.
The target answer is provided only to help you align your reasoning;
you must not copy the answer, quote it, or mention that the answer is known.

Your output must contain ONLY the completed and optimized CoT text.
No explanation, no meta-commentary, no references to the instructions.

[Instruction]
### Task Data

**Question:**
{question}

**Target Answer (For your internal guidance ONLY - Do NOT leak):**
{answer}

**Draft CoT (Start with this, complete it, and refine):**
{cot}
---
Based on the Question and Target Answer, please output the complete, optimized CoT.
It must start with the logic in the Draft CoT, flow naturally to the Target Answer, and strictly avoid redundancy.

---

*Figure 9.* Complete prompt for MSC refinement. The prompt guides the refinement model to enhance logical completeness and conciseness while maintaining stylistic consistency with the original reasoning trajectory.

## C.2. Refinement Examples

Figures 10 and 11 illustrate concrete examples comparing raw MSC and refined MSC.

**Question:** The sum of two numbers, A and B, is 1149. When A is divided by B, the quotient is 8 and the remainder is 24. Therefore, A is ?

**Ground Truth:** 1024

**[Raw MSC]**

Okay, let's see here. The problem says that the sum of two numbers, A and B, is 1149. When A is divided by B, the quotient is 8 and the remainder is 24. We need to find the value of A. Hmm, alright. Let me try to break this down step by step.

First, I know that when you divide one number by another, you get a quotient and a remainder. The formula for that is usually something like Dividend = Divisor × Quotient + Remainder. So, in this case, A divided by B gives a quotient of 8 and a remainder of 24. That should translate to A = B × 8 + 24. Let me write that down:
A = 8B + 24.

Also, the sum of A and B is 1149. So, A + B = 1149. Got that. Now, since we have two equations here, maybe we can substitute one into the other. Since we already have A expressed in terms of B from the first equation, we can plug that into the second equation.

So substituting A in the sum equation:
(8B + 24) + B = 1149.

Combine like terms here. 8B + B is 9B, so:
9B + 24 = 1149.9B + 24 = 1149.

**[Refined MSC]**

Okay, let's see here. The problem says that the sum of two numbers, A and B, is 1149. When A is divided by B, the quotient is 8 and the remainder is 24. We need to find the value of A. Hmm, alright. Let me try to break this down step by step.

First, I know that when you divide one number by another, you get a quotient and a remainder. The formula for that is usually something like Dividend = Divisor × Quotient + Remainder. So, in this case, A divided by B gives a quotient of 8 and a remainder of 24. That should translate to A = B × 8 + 24. Let me write that down:
A = 8B + 24.

Also, the sum of A and B is 1149. So, A + B = 1149. Got that. Now, since we have two equations here, maybe we can substitute one into the other. Since we already have A expressed in terms of B from the first equation, we can plug that into the second equation.

So substituting A in the sum equation:
(8B + 24) + B = 1149.

Combine like terms here. 8B + B is 9B, so:
9B + 24 = 1149.

Subtract 24 from both sides:
9B = 1125.

Now divide both sides by 9:
B = 125.

Now plug that back into the expression for A:
A = 8 × 125 + 24 = 1000 + 24 = 1024.

Check: 1024 + 125 = 1149 — correct. And 1024 ÷ 125 = 8 with remainder 24 — also correct.
So A is 1024.

*Figure 10.* Refinement example demonstrating logical completion. Raw MSC stops mid-reasoning; refined MSC completes the derivation while preserving the original flow.

**Question:** Ana can run a circular track three times in 8 minutes. Maria can run the same track twice in 5 minutes. If Ana and Maria start running at the same time from the start, what is the total number of laps they will run before they meet again at the start for the first time? How many times has each of them run the track when they meet again at the start for the first time?

**Ground Truth: 31**

**[Raw MSC]**

Okay, so the problem is about Ana and Maria running around a circular track. Ana can do three laps in 8 minutes, and Maria can do two laps in 5 minutes. They start at the same time and place, and we need to figure out after how much time they'll meet again at the start for the first time. Then, we have to find out how many laps each has run by that time.

First, let's break down the information given. For Ana, three laps take 8 minutes. So, her speed is 3 laps per 8 minutes. For Maria, two laps take 5 minutes, so her speed is 2 laps per 5 minutes. We need to find the time when both of them have completed an integer number of laps, meaning that the time must be a multiple of both Ana's lap time and Maria's lap time. That sounds like a least common multiple (LCM) problem. But maybe I need to convert their speeds into rates or find their individual lap times.

Let me think. If Ana does three laps in 8 minutes, her time per lap is 8/3 minutes per lap. Similarly, Maria's time per lap is 5/2 minutes per lap. So, we need to find the LCM of 8/3 and 5/2. Hmm, LCM for fractions? I remember that LCM for fractions is calculated by taking LCM of the numerators divided by GCD of the denominators. Wait, let me verify that.

Actually, another approach: the time when they meet again must be a multiple of both their lap times. So, if Ana takes 8/3 minutes per lap, then her time to do 'm' laps is (8/3)*m minutes. Similarly, Maria's time for 'n' laps is (5/2)*n minutes. We need these two times to be equal for some integers m and n. So, (8/3)*m = (5/2)*n. Then, solving for m/n gives (5/2)/(8/3) = (5/2)*(3/8) = 15/16. So m/n = 15/16. Therefore, the smallest integers m and n that satisfy this are m=15 and n=16. Then, substituting back into the time, (8/3)*15 = 40 minutes, and (5/2)*16 = 40 minutes. So after 40 minutes, they both meet again at the start. Then, the total number of laps they will have run is 15 + 16 = 31 laps.

**[Refined MSC]**

Okay, so the problem is about Ana and Maria running around a circular track. Ana can do three laps in 8 minutes, and Maria can do two laps in 5 minutes. They start at the same time and place, and we need to figure out after how much time they'll meet again at the start for the first time. Then, we have to find out how many laps each has run by that time.

First, let's break down the information given. For Ana, three laps take 8 minutes. So, her time per lap is 8/3 minutes. For Maria, two laps take 5 minutes, so her time per lap is 5/2 minutes. We need to find the smallest time at which both have completed an integer number of laps — meaning the time must be a common multiple of their individual lap times.

To find the least common multiple of two fractions, we can use the formula: LCM(a/b, c/d) = LCM(a,c) / GCD(b,d). But a more straightforward way is to find the smallest time T such that T is divisible by both 8/3 and 5/2. That means T = (8/3) * m = (5/2) * n for some positive integers m and n.

Setting the two expressions equal: (8/3)m = (5/2)n.
Cross-multiplying: 16m = 15n.
So m/n = 15/16.
The smallest integers satisfying this are m = 15 and n = 16.

Now plug back in to find T:
Ana's time: (8/3) * 15 = 40 minutes.
Maria's time: (5/2) * 16 = 40 minutes.

So they meet again at the start after 40 minutes.
Ana has run 15 laps, Maria has run 16 laps.
Total laps: 15 + 16 = 31.

*Figure 11.* Refinement example: reasoning optimization. Raw MSC contains exploratory backtracking; refined MSC eliminates redundancy while maintaining the core logic.

