# OpenReview forum: "SuCo: Sufficiency-guided Continuous Adaptive Reasoning"
_ICML.cc/2026/Conference — ICML 2026 regular_

### Official Review · Reviewer_vry9 · 2026-03-03

**Soundness:** 2
**Presentation:** 3
**Significance:** 2
**Originality:** 3
**Overall Recommendation:** 4
**Confidence:** 4

**Summary:**

This paper addresses the inefficiency of current reasoning models, which generate excessively long and costly reasoning chains even for simple queries, lacking a principled criterion to stop thinking. To solve this, the authors propose SuCo, a continuous adaptive reasoning framework with a two-stage training pipeline.

**Compliance With Llm Reviewing Policy:**

Affirmed.

**Final Justification:**

The authors addressed my concerns during the rebuttal period and provided additional experiments, which made the work more complete. Therefore, I have revised my score from 3 to 4.

**Key Questions For Authors:**

See Weaknesses.

**Limitations:**

Yes.

**Strengths And Weaknesses:**

Strengths:
1. The paper introduces an effective pipeline combining Supervised Fine-Tuning (SFT) and Reinforcement Learning (RL). The ablation studies clearly prove the necessity of this dual approach.
2. Evaluated across math, code, and science, SuCo achieves massive efficiency gains while consistently outperforming full-CoT baselines in accuracy.

Weaknesses:
1. The training and test sets share the exact same domains. To prove true generalization rather than domain-specific length memorization, can SuCo autonomously adapt its reasoning length on completely unseen domains?
2. The refinement step uses a powerful 80B LLM to eliminate redundancy and complete logical gaps. This weakens the necessity of the complex mathematical MSC truncation. How does this pipeline compare to a baseline that simply prompts the 80B LLM to compress the full CoT directly? Furthermore, when the LLM generates new text to bridge logical gaps during refinement, how do you theoretically or empirically guarantee it does not re-introduce the redundant reasoning the method aims to eliminate?
3. Estimating problem complexity via length percentiles across the entire training set seems biased. Different domains have inherently different baseline lengths (e.g., code generation naturally requires more tokens than a simple math proof). How do you ensure this cross-domain percentile calculation remains an objective measure of difficulty?
4. Figure 6 shows SuCo adapts its reasoning length to problem difficulty, but base LRM models inherently do this as well (harder problems naturally induce longer generations). Adding the pre-trained baseline model's length-to-difficulty curve to Figure 6 is necessary to isolate and demonstrate the specific marginal effectiveness of SuCo's adaptive training.

---

> ### Author Rebuttal · Authors · 2026-03-30
>
> Thank the reviewer for the insightful comments. Below we respond to each concern and hope that our clarifications are helpful. If our response resolves your concerns, we would greatly appreciate it if you could raise your score (3: Weak reject). We are happy to further clarify any remaining issues.
>
> ---
> > Q1. Generalization
>
> To assess whether SuCo truly generalizes, rather than merely memorizing domain-specific reasoning lengths from the training set, we conducted out-of-domain evaluations.
>
> |Method|StrategyQA (ACC / Tokens)|CommonsenseQA (ACC / Tokens)|AlpacaEval2.0 (LC Win Rate / Tokens)|
> |-------------------|-----------------------|--------------------------|----------------------------------|
> |DeepSeek-R1-Distill|53.3/483|45/743|1.05/596|
> |Full CoT SFT|22.6/742|19.4/1,061|0.3/743|
> |SuCo|55.7/442|49.3/369|2.4/288|
>
> These results indicate that **the adaptive behavior learned by SuCo transfers beyond the original training domains**.
>
> ---
> > Q2. Role of the 80B refinement model
>
> We compares our method against direct compression and a weaker refinement model.
> |Method|Train CoT Tokens|Accuracy (%)|Inference Tokens|
> |--------------------|----------------|-----------|----------------|
> |Full CoT SFT|3,781|38.4|5,082|
> |Direct Compress (80B)|2,519|38.7|3,917|
> |MSC w/o srefine|1,803|43.5|2,189|
> |MSC w/ refine (7B)|1,441|44.7|1,622|
> |MSC w/ refine (80B)|1,138|45.7|1,344|
>
> These results show that **MSC truncation is the main source of improvement，while the 80B refinement is only a complementary gain.**
>
> The key role of MSC is to identify the shortest prefix along the original high-quality reasoning path, providing a much stronger structure prior than post-hoc compression. In contrast, direct compression has no notion of sufficiency and does not naturally support adaptive reasoning control. Importantly, 40.7% of MSC samples are empty-think, which direct compression cannot recover.
>
> For the concern that refinement may re-introduce redundancy,  the examples in Appendix C confirm that refinement operates as local patching, not large-scale rewriting. Quantitatively, refinement reduces average tokens from 1,803 to 1,138 (a 37% reduction), leading the net effect is consistently compressive.
>
> ---
> > Q3. Cross-domain complexity estimation bias
>
> We sincerely thank the reviewer for this insightful critique. We agree that baseline lengths vary across domains. However, we argue **this bias has limited impact on our results, and SuCo remains robust under this setting**.
>
> (1) Although percentiles are computed globally, they **preserve monotonicity within each domain**. That is, within a given domain (e.g., math), more difficult problems still correspond to higher percentiles. Therefore, the percentile still functions as a meaningful surrogate difficulty signal rather than an absolute length threshold.
>
> (2) To further validate this concern, we conducted an additional experiment using Intra-domain Percentile.
>
> |Method|Math||Code||Science||Avg.||
> |-----------------------------|----|------|----|------|-------|------|----|------|
> ||acc|tokens|acc|tokens|acc|tokens|acc|tokens|
> |Full CoT SFT|59.1|5,380|28.5|6,345|27.7|3,522|38.4|5,082|
> |MFT (Cross-domain Percentile)|69.1|1,359|33.2|1,706|34.8|966|45.7|1,344|
> |MFT (Intra-domain Percentile)|69.2|1,376|33.2|1,692|35.1|1,021|45.8|1,363|
>
> As shown above, **Intra-domain yields nearly identical performance to the Cross-domain setting**. This indicates that **SuCo is robust to cross-domain percentile estimation**, and the global percentile remains a reliable and objective measure of reasoning difficulty in practice.
>
>
> ---
> > Q4. base LRMs also naturally adapts
>
> We thank the reviewer for this important suggestion and have added the [base LRM (DeepSeek-R1-Distill-Qwen) length–difficulty curve](https://anonymous.4open.science/r/asdfg-413E/figure1.png).
>
> As expected, the base model also scales reasoning length with difficulty. However, the two models differ in two important respects.
> (1) SuCo exhibits a higher difficulty-sensitivity ratio (L5/L1 $\approx$ 5.5×) compared to the base LRM ($\approx$3.1×), indicating that **SuCo's length adaptation is more discriminative across difficulty levels**.
> (2) SuCo operates at a fundamentally different efficiency regime: it uses 89% fewer tokens on easy problems while still improving accuracy, whereas the base LRM expends substantial computation even on trivial queries. This suggests that the base LRM's length variation reflects an inability to truncate unnecessary reasoning, while **SuCo's variation reflects genuine difficulty-conditioned allocation learned through sufficiency-aware training**.

---

> > ### Author Rebuttal · Reviewer_vry9 · 2026-04-02
> >
> > Thank you for the authors’ rebuttal and the additional experiments. I appreciate the effort to strengthen the paper, and in my view, the other concerns have been largely addressed. If this remaining point can also be clarified, I would be happy to reconsider my score.
> >
> > For the OOD evaluation, it would be very helpful to include comparisons with all of the baselines considered in Table 1 (e.g., AdaCoT), as this would make the OOD generalization claim more convincing. If a full comparison on all OOD benchmarks is not feasible during the discussion period, could the authors provide such a comparison on at least one representative OOD dataset?

---

> > > ### Author Response · Authors · 2026-04-02
> > >
> > > > Q. Full baseline comparison on OOD benchmarks
> > >
> > > We thank the reviewer for the follow-up and for emphasizing the importance of OOD evaluation.
> > >
> > > To address this, we run all baselines on all three OOD benchmarks.
> > >
> > > | Method              | StrategyQA (ACC / Tokens) | CommonsenseQA (ACC / Tokens) | AlpacaEval2.0 (LC Win Rate / Tokens) |
> > > | ------------------- | ------------------------- | ---------------------------- | ------------------------------------ |
> > > | DeepSeek-R1-Distill | 53.3 / 483                | 45.0 / 743                     | 1.05 / 596                           |
> > > | Full CoT SFT        | 22.6 / 742                | 19.4 / 1,061                 | 0.3 / 743                            |
> > > | AdaCoT              | 54.2 / 464                | 46.1 / 573                   | 1.48 / 331                           |
> > > | AdaptThink          | 53.6 / 512                | 44.7 / 647                   | 0.78 / 429                           |
> > > | S-GRPO              | 54.1 / 422                | 47.2 / 529                   | 1.87 / 321                           |
> > > | LHRMs               | 52.8 / 382                | 46.8 / 441                   | 1.54 / 297                           |
> > > | SuCo (Ours)         | 55.7 / 442                | 49.3 / 369                   | 2.4 / 288                            |
> > >
> > > As shown in table, across all three out-of-domain benchmarks, SuCo achieves the highest accuracy, while maintaining competitive or best-in-class token efficiency. These results demonstrate that SuCo's adaptive reasoning behavior reflects genuine generalization rather than domain-specific length memorization.
> > >
> > > We hope this addresses the reviewer’s concern and would appreciate a reconsideration of the score.

---

### Official Review · Reviewer_nDG6 · 2026-03-10

**Soundness:** 2
**Presentation:** 3
**Significance:** 2
**Originality:** 2
**Overall Recommendation:** 4
**Confidence:** 4

**Summary:**

This paper addresses the computational inefficiency of Large Reasoning Models (LRMs) generating excessively long Chain-of-Thought (CoT) trajectories. The authors introduce "Minimal Sufficient CoT" (MSC), the shortest reasoning prefix required to confidently produce a correct answer. Building on MSC, they propose SuCo, a two-stage framework enabling models to autonomously adjust reasoning length. Stage 1 (MFT) fine-tunes the model on curated MSC data using problem-adaptive difficulty thresholds. Stage 2 (SAPO) optimizes the model via reinforcement learning, utilizing a dynamic complexity pool and a custom reward function to penalize both over-thinking and under-thinking. Empirically, SuCo significantly reduces token consumption while improving accuracy across math, code, and science benchmarks.

**Compliance With Llm Reviewing Policy:**

Affirmed.

**Final Justification:**

The authors' response has addressed my initial questions, and I am happy to raise my rating.

**Key Questions For Authors:**

See weakness.

**Limitations:**

Yes.

**Strengths And Weaknesses:**

**Strengths**

* The paper features highly aesthetically pleasing charts and overall layout.
* The presentation is exceptionally clear, and the mathematical formulations are standard and rigorous.

**Weaknesses**

* The definition of "sufficiency" lacks clear motivation and corresponding experimental validation. The authors should justify why it is defined this way and discuss how it compares to alternative formulations. Furthermore, in code generation tasks (such as MBPP and LiveCodeBench used in the evaluation), the target answer $y^*$ can be a lengthy code snippet. In such cases, the joint probability or geometric mean of a long sequence becomes extremely fragile, and answer lengths vary drastically across different problems. Using a  $\delta$ or a percentile-based estimation for thresholds will inevitably introduce massive variance in long-answer tasks.
* The MSC refinement step relies on a powerful external 80B model (Qwen3-Next-80B-A3B-Instruct) to bridge logical gaps and eliminate redundancy. This raises a critical question: is the ultimate performance gain genuinely attributed to the proposed "Minimal Sufficient CoT (MSC)" threshold truncation, or is it merely the result of the 80B teacher model conducting high-quality logical rewriting and data distillation? The authors should clarify this by testing whether the performance improvements still hold if the refinement model is replaced by the base model itself.
* The proposed method introduces an excessive number of hyperparameters, such as the base threshold $\delta_0$, sensitivity coefficient $\alpha$, minimum reasoning length $L_{min}$, sufficiency reward weight $\beta$, over-thinking penalty coefficient $\lambda_{over}$, and under-thinking penalty coefficient $\lambda_{under}$. This heavy reliance on precise hyperparameter tuning raises significant doubts regarding the method's generalizability.
* The paper fails to evaluate on commonly used metrics such as pass@k.
* The manuscript lacks discussion on several highly relevant prior works. The following papers specifically address the phenomena of overthinking and underthinking in reasoning models, and the authors should consider mentioning them:
  * Thinkprune: Pruning long chain-of-thought of llms via reinforcement learning.
  * Thoughts Are All Over the Place: On the Underthinking of o1-Like LLMs
  * CyclicReflex: Improving Reasoning Models via Cyclical Reflection Token Scheduling.
  * Alphaone: Reasoning models thinking slow and fast at test time.
  * Does Thinking More Always Help? Mirage of Test-Time Scaling in Reasoning Models.

---

> ### Author Rebuttal · Authors · 2026-03-30
>
> Thank the reviewer for the insightful and constructive feedback. If our response resolves your concerns, we would greatly appreciate it if you could raise your score (3: Weak reject). We are happy to clarify any remaining issues.
>
> ---
> > Q1. The definition of sufficiency
>
> (1) Motivation of the Definition
>
> The objective of MSC is to evaluate whether a reasoning prefix $z$ is adequate to derive the answer $y^* $. Therefore, the most intuitional signal for sufficiency is the conditional probability $P(y^* \mid x, z)$. This formulation directly anchors "reasoning sufficiency" to "answer supportability," providing a verifiable and task-agnostic metric.
>
> **To address the "fragility" of long sequences, we employ the geometric mean**:
> $$
> \mathcal{S} =  ( \prod_{i=1}^{||y^* ||} \pi_\theta(y_i^* \mid x, z, y_{<i}^* )^{1/||y^* ||} \implies \log \mathcal{S} = \frac{1}{||y^* ||} \sum_{i=1}^{||y^* ||} \log \pi_\theta(y_i^* \mid x, z, y_{<i}^* )
> $$
> By taking the $1/||y^* ||$ root, we normalize the joint probability into token-level average probability. Unlike joint probability, which decays exponentially as sequence length increases, the geometric mean remains stable with varying answer lengths.
>
> (2) Experimental Validation
>
> We compared the geometric mean against alternative formulations.
>
> |Method|Train CoT Tokens|Accuracy (%)|Inference Tokens|
> |---|---|---|---|
> |Full CoT SFT|3,781|38.4|5,082|
> |Joint Probability|2,149|40.1|2,573|
> |Arithmetic Mean|1,341|43.1|1,578|
> |GeometricMean|1,138|45.7|1,578|
>
> These results demonstrate that **geometric mean provides a more effective sufficiency signal**.
> We will add this motivation and the above ablation to the revised paper for clarity.
>
> ---
> > Q2. Contribution of MSC vs. 80B refinement
>
> MSC is the key mechanism that enables **adaptive reasoning control**. In contrast, **refinement merely plays the role of post-processing**: raw truncation may leave incomplete transitions, and refinement makes them more fluent, rather than introducing a new reasoning strategy. This is also consistent with the examples in Appendix C, where the edits mainly polish abrupt boundaries and remove redundancy.
>
> We compares our method against direct compression and a weaker refinement model:
> |Method|Train CoT Tokens|Accuracy (%)|Inference Tokens|
> |--------------------|----------------|-----------|----------------|
> |Full CoT SFT|3,781|38.4|5,082|
> |Direct Compress (80B)|2,519|38.7|3,917|
> |MSC w/o srefine|1,803|43.5|2,189|
> |MSC w/ refine (7B)|1,441|44.7|1,622|
> |MSC w/ refine (80B)|1,138|45.7|1,344|
>
> These results support three clear conclusions.
>
> - **MSC truncation alone already provides a large gain**.
> - **Refinement is helpful but not the main driver**. Adding 7B refinement improves performance by 1.2%, and 80B refinement provides a further 1.0% gain. In contrast, MSC selection itself yields a much larger improvement over Direct Compression (+4.8%).
> - **High-quality rewriting alone is not sufficient**. If the gain mainly came from the 80B teacher rewriting trajectories, then Direct Compress (80B) should be competitive. However, it is far below MSC w/o refine.
>
> This shows that **MSC truncation is much more important than simply compressing the full CoT**.
>
> ---
> > Q3. An excessive number of hyperparameters
>
> **The key hyperparameters are well-justified and robust, not precision-tuned.**
>
> The ablations on $L_{\min}$, EMA rate $\eta$, and over-thinking tolerance  $\epsilon$ (Appendix A) show that the method remains **effective over broad parameter ranges**, rather than at a narrowly tuned optimum.
>
> **Moreover, not all coefficients require independent tuning**. $\beta$ is the standard reward weight used in RL practice, and $\lambda_{\text{over}}$ and $\lambda_{\text{under}}$ are simply set symmetrically to 0.5, treating them with equal importance.
>
> For MSC construction, $\delta_0=0.5$ and $\alpha=0.4$ yields an average threshold of ~0.7, which is aligned with the best static threshold in Table 2. This choice is therefore well motivated.
>
> Overall, SuCo is robust rather than fragile. Its gains come from the sufficiency-guided design itself, not from extreme hyperparameter optimization.
>
> ---
> > Q4. Fails to evaluate on pass@k
>
> We originally reported pass@1 because most related works evaluate using pass@1. Following the reviewer’s suggestion, we additionally report pass@5 and pass@10.
>
> |Method|Pass@1|Pass@5|Pass@10|
> |-------------|--------|--------|--------|
> |Math-Base|14.1|17.4|18.6|
> |Math-Instruct|32.8|36.2|37.7|
> |Full|38.4|43.9|45.1|
> |AdaCoT|47.2|53.7|55.4|
> |AdaptThink|47.8|53.6|56.3|
> |S-GRPO|49.2|55.7|58.5|
> |LHRMs|50.5|57.0|60.2|
> |SuCo|53.1|60.1|63.7|
>
> As shown, SuCo consistently improves performance across all pass@k metrics.
>
> ---
> > Q5. Lacks discussion on several works
>
> We thank the reviewer for pointing out these highly relevant works. **We will discuss them in the Related Work section.**

---

> > ### Author Rebuttal · Reviewer_nDG6 · 2026-04-03
> >
> > Thank you for the clarification. The authors’ response has resolved my questions, and I will be raising my rating accordingly.

---

> > > ### Author Response · Authors · 2026-04-04
> > >
> > > Thank the reviewer for the thorough review and constructive suggestions. We are glad that our responses have addressed all your concerns.

---

### Official Review · Reviewer_wKAm · 2026-03-12

**Soundness:** 4
**Presentation:** 2
**Significance:** 3
**Originality:** 3
**Overall Recommendation:** 4
**Confidence:** 3

**Summary:**

This proposes SuCo, a training framework that controls LLM reasoning length to achieve minimum sufficient CoT (MSC). The authors first formulate MSC in a probabilistic view as the shortest reasoning prefix satisfying sufficiency and design a schema for adaptive complexity control. Then they introduce their two-stage training framework as first learning by distillation from a strong LRM followed by RL fine-tuning with a sufficiency-aware reward. The experiment results show that SuCo achieves the best performance and reasoning length control  compared to other full CoT and adaptive LRM baselines, and validate the importance of components within SuCo.

**Compliance With Llm Reviewing Policy:**

Affirmed.

**Final Justification:**

I thank the authors for their effort in rebuttal and I will maintain my positive score.

**Key Questions For Authors:**

1. The usage of some notations is not appropriate:
- As defined in Section 3.1, $\|\|z\|\|$ denotes the length of a specific CoT trace, and then in Section 3.4, $\|\|z_i^{avg}\|\|$ is used to denote an evolving reasoning length that does not correspond to any specific CoT trace. Simultaneously the former definition applies to $\|\|z_i^k\|\|$. This can make $\|\|z_i^{avg}\|\|$ confusing to readers, so I suggest using a separate notation for this variable in Section 3.4.
- Above Equation 9, the authors wrote "The pool is initialized from $\pi_{MFT}$" without explicitly defining $\pi_{MFT}$. I guess it refers to the policy after SFT, and it should be explicitly stated.

2. I am a bit confused with **the motivation of the Problem-Adaptive Threshold**, although I am convinced that it is practically effective as in the experiments. MSC is defined as the minimal prefix that enables the LLM policy to provide a correct answer at a threshold probability in token-level average. If the query is more complex itself, even for a fixed $\delta$ the difficulty becomes higher, then why is the additional complexity term necessary? I feel more clarification on this point can make this part more convincing and well motivated.

3. In the **Baselines** paragraph in Section 4.1, the author mentioned that "*For fair comparison, AdaCoT and LHRMs are initialized from Qwen2.5-Math-Base and trained on the same source data as SuCo*". However, it is not clear to the audience what the "source data" refers to. Does it refer to the MFT data or the vanilla CoT data on the same dataset? This should be specified.

4. SuCo relies heavily on $M_{LRM}$ for cold-start data collection and initial complexity estimation. I wonder how the performance will be if there is no MFT phase and the policy is trained from scratch with SAPO.

5. Continuing on Q3, although the empirical improvement of SuCo over baselines is consistent, given that:
- In Table 2, the performance of MFT is sensitive to some design choices
- In Table 3, the performance of SAPO is much more stable even without DCP or $R_{suff}$, and MFT already achieves a better performance against baselines.

Readers may wonder the importance of SAPO and its design choices in SuCo. I encourage the authors to provide more evidence, either quantitatively or qualitatively, to justify that SAPO enables some patterns that simple distillation cannot achieve. Figure 5 may be a good starting point.

**Limitations:**

No. I would encourage the authors to discuss two main points as limitations and future works:

1. The reliance on distillation from strong LRMs.

2. Potential extension of SuCo to agentic tasks.

**Strengths And Weaknesses:**

**Strengths**

1. The paper is well motivated and the efficiency of long CoT is important for applications of LLMs.

2. The formulation of sufficiency is interesting and makes sense to me.

3. SuCo shows strong empirical performance across various datasets and the ablation study is generally well implemented.

**Weaknesses**

1. The MFT dataset and dynamic complexity estimation relies heavily on data from a strong LRM.

2. Although the overall idea is clearly explained, some parts of presentation can be improved for better readability. See Q1-3 below.

3. The effectiveness of SAPO over MFT is not significant to me. See Q4 and Q5 below.

---

> ### Author Rebuttal · Authors · 2026-03-30
>
> Thank the reviewer for recognizing the contributions of our work and for the valuable feedback. We respond to each comment as follows and sincerely hope that our rebuttal could properly address your concerns. If the concerns are resolved, we would deeply appreciate it if you could raise your score (4: Weak accept). We remain happy to address any remaining issues.
>
> ---
>
> > Q1. Notation and clarity
> >
> > $||z_i^{avg}||$, $\pi_{MFT}$, “source data”
>
> We appreciate the reviewer’s meticulous suggestions and will revise these points to improve clarity and readability.
>
> In the revision, we will (1) replace $||z_i^{avg}||$ in Section 3.4 with a distinct notation to avoid confusion; (2) explicitly state that $\pi_{MFT}$ refers to the policy obtained after the Stage I (MFT) training; and (3) clarify in Sec. 4.1 that the “source data” refers to the vanilla CoT data.
>
> ---
>
> > Q2. Motivation of Adaptive Threshold
> >
> > If the query itself is more complex, then even with a fixed $\delta$, the difficulty is already reflected. Why is the additional complexity term necessary?
>
> We thank the reviewer for this insightful question.
>
> A fixed $\delta$ governs the confidence bar uniformly across all problems, therefore fails to calibrate the expected sufficiency level to problem difficulty.
>
> In contrast, the adaptive threshold lowers the bar for simple problems to allow shorter MSCs, while raises it for hard problems to avoid premature truncation. **This produces a more discriminative MSC distribution across difficulty levels, which in turn provides a stronger adaptive prior for subsequent training.**
>
> This design also carries through to the SAPO stage. The dynamic complexity pool continuously updates $\delta$ as the policy evolves, **providing real-time sufficiency signals and better aligning reasoning length with problem complexity**.
>
> Empirically, this motivation is supported by the threshold ablations in Table 2 and the DCP ablation in Table 3.
>
> ---
>
> > Q3. Importance of SAPO over MFT
>
> While MFT provides a strong initialization by distilling static MSC labels, **SAPO is crucial for transforming the model from "label-fitting" to "dynamic problem-solving."**
>
> (1) We conducted an additional ablation initializing SAPO directly from DeepSeek-R1-Distill.
>
> |**Starting Point**|**Method**|**Accuracy (%)**|**Tokens**|
> |-------------------|-------------|----------------|----------|
> |DeepSeek-R1-Distill|Baseline|45.2|5,736|
> |DeepSeek-R1-Distill|+SAPO|50.7|2,134|
> |MFT|Baseline|51.5|1,347|
> |MFT|+SAPO (SuCo)|53.1|1,483|
>
> As shown in the table, **SAPO provides stable and consistent gains.**
>
> (2) **SAPO reshapes the reasoning length distribution.**
>
> MFT learns to replicate MSC lengths estimated from a static offline dataset. These estimates are inherently fixed and cannot adapt to individual questions. As shown in Figure 5 of the paper, SAPO actively recalibrates per-problem reasoning effort: on simple problems it compresses reasoning, while on hard problems it further allocates additional budget when needed. This is a behavioral pattern that SFT alone cannot produce.
>
> (3) **SAPO generalizes to unseen domains.**
>
> |Method|StrategyQA (ACC / Tokens)|CommonsenseQA (ACC / Tokens)|AlpacaEval2.0 (LC Win Rate / Tokens)|
> |-------------------|-----------------------|--------------------------|----------------------------------|
> |DeepSeek-R1-Distill|53.3/483|45/743|1.05/596|
> |Full CoT SFT|22.6/742|19.4/1,061|0.3/743|
> |MFT|28/213|26.6/342|0.67/314|
> |SuCo|55.7/442|49.3/369|2.4/288|
>
> MFT overfits MSC patterns from the training distribution and cannot generalize sufficiency judgments to novel query types. By contrast, SuCo substantially outperforms even the DeepSeek-R1-Distill baseline on all three tasks while using fewer tokens. This demonstrates that SAPO learns a generalizable policy for how to calibrate reasoning, rather than memorizing domain-specific length patterns from training data.
>
> ---
>
> > Q4. Limitation. The reliance on distillation from strong LRMs.
>
> We will add this as a limitation. Importantly, our contribution lies in the MSC-based supervision and sufficiency-aware optimization, rather than in the teacher model itself. As shown in Table 2, removing the 80B refinement reduces average accuracy while still outperforming all baselines.
>
> ---
>
> > Q5. Potential extension of SuCo to agentic tasks.
>
> We thank the reviewer for identifying these valuable paths for future research.
>
> In agentic scenarios, "sufficiency" is even more critical, as over-thinking can lead to redundant API costs, while under-thinking leads to task failure. Extending SuCo to agentic tasks is a compelling but promising direction.

---

> > ### Author Rebuttal · Reviewer_wKAm · 2026-04-02
> >
> > I thank the authors for their effort in this thorough rebuttal and my concerns have been fully addressed. I will maintain my current score. Good luck with your submission.

---

> > > ### Author Response · Authors · 2026-04-02
> > >
> > > Thank the reviewer for the thorough review and constructive suggestions. We are glad that our responses have addressed all your concerns.

---

### Official Review · Reviewer_fBsN · 2026-03-13

**Soundness:** 3
**Presentation:** 3
**Significance:** 3
**Originality:** 3
**Overall Recommendation:** 4
**Confidence:** 4

**Summary:**

This work first introduces Minimal Sufficient CoT (MSC) to represent the shortest CoT prefix that can lead to the correct answer by measuring the probability of the correct answer given a partial sequence. Then it uses sufficiency-based SFT and RL to train the model to adaptively control the reasoning effort based on varying difficulty. Experiments on math, code, and science benchmarks show the effectiveness and efficiency of SuCo.

**Compliance With Llm Reviewing Policy:**

Affirmed.

**Key Questions For Authors:**

1. How many GPU hours does the training take in total, and how does this compare to the baselines?
2. Given that $S_\theta$ depends on the base model's probability distribution, how consistent are MSC boundaries across different base models? Does the model-dependent nature of MSC labels limit the transferability of the constructed datasets to other model families?

**Limitations:**

No. The paper does not adequately discuss its limitations such as the restriction to closed-form tasks due to the ground-truth-dependent sufficiency score and the model-dependent nature of MSC labels.

**Strengths And Weaknesses:**

**Strengths**
1. The idea of using minimal sufficient CoT is interesting and well-formalized.
2. The two-stage design is sound, and the EMA-based update mechanism is a thoughtful contribution.
3. The results are strong, with clear evidence of continuous adaptivity in reasoning behavior across difficulty levels.

**Weaknesses**
1. The overall pipeline is relatively complex. It would be helpful to discuss how sensitive the method is to model calibration, since the sufficiency score relies directly on the model's own token probabilities.
2. The sufficiency measure requires access to the ground-truth answer at construction time, which limits applicability to tasks with verifiable, closed-form answers. It is unclear how the method would generalize to open-ended tasks or problems with multiple valid answers.

---

> ### Author Rebuttal · Authors · 2026-03-30
>
> Thank the reviewer for the insightful and constructive comments. Below we address each concern and will incorporate the clarifications and additional results in the revision. If our response resolves your concerns, we would greatly appreciate it if you could raise your score (4: Weak accept). We are happy to further clarify any remaining issues.
>
> ---
>
> > Q1. Calibration sensitivity
> >
> > How sensitive is the method to model calibration?
> >
> > How consistent are MSC boundaries across different base models?
>
> This is a well-motivated concern, but **our results suggest that SuCo is robust to calibration variance and demonstrates strong cross-model transferability**.
>
> From a per-sample perspective, different calibrator models can indeed produce slightly different MSC boundaries. However, such variations are typically small rather than a change in the core reasoning content. Moreover, the refinement process completes truncated transitions and normalizes stylistic inconsistencies, which further reduces the effect of such boundary variation.
>
> At the level of dataset, **what matters for training is the ordinal structure**. In practice, easy problems still tend to yield shorter sufficient traces and harder problems longer ones across different calibrator models, so the overall supervision signal remains stable.
>
> More importantly, our objective is not to define an absolute, model-agnostic “unique shortest reasoning.” Instead, MSC serves as a practical supervision target that is more concise than full CoT while still sufficient to reach the correct answer.
>
> In addition, the second-stage SAPO training further adapts the policy using the target model’s own sufficiency signal, which helps absorb residual mismatch from offline MSC labels.
>
> We further validated this point empirically by constructing MSC data using different calibrator models and training different target models.
>
> | Annotation Model|Qwen2.5-1.5B|Llama-3.2-3B|
> | ---------------------------|------------|------------|
> | Full CoT SFT (Baseline)|37.4(5,177)|38.1(5,084)|
> | Qwen3-4B|44.7(1,394)|44.2 (1,524) |
> | Qwen3-14B|44.2(1,521)|43.2(1,821)|
> | DeepSeek-R1-Distill-Qwen-7B|44.5(1,491)|43.7(1,691)|
>
> As shown above, across all settings, MSC supervision derived from different calibrators consistently outperforms full-CoT training. These results suggest that **MSC boundaries are robust across models and that the constructed datasets transfer well across model families**.
>
> ---
>
> > Q2. Generalization to Open-ended Tasks
>
> We thank the reviewer for raising this important question. We agree that SuCo relies on ground-truth answers to compute the sufficiency score, and will discuss it in the limitations section.
>
> However, this requirement applies only during offline dataset construction, not at inference time. **Once training is complete, the model has internalized adaptive reasoning as a general capability**.
>
> Besides, overthinking in LRMs is most pronounced on closed-form tasks like math, competitive programming, and scientific QA.  Therefore, **SuCo already addresses a practically important regime**.
>
> To further assess generalization, we conducted OOD experiments including open-ended tasks. We observe that SuCo still maintains improved accuracy while reducing reasoning length compared to full-CoT baselines. This suggests that **the learned adaptive reasoning capability generalizes beyond closed-form tasks**.
>
> |Method|StrategyQA (ACC / Tokens)|CommonsenseQA (ACC / Tokens)|AlpacaEval2.0 (LC Win Rate / Tokens)|
> |-------------------|-----------------------|--------------------------|----------------------------------|
> |DeepSeek-R1-Distill|53.3/483|45/743|1.05/596|
> |Full CoT SFT|22.6/742|19.4/1,061|0.3/743|
> |SuCo|55.7/442|49.3/369|2.4/288|
>
> Finally, we agree that **extending sufficiency estimation to open-ended settings is an important and promising future direction**. We will include this in the revised paper.
>
> ---
>
> > Q3. GPU hours
> >
> > How many GPU hours does the training take in total, and how does this compare to the baselines?
>
> We report the total training cost in GPU hours using 8 × NVIDIA H100 80GB GPUs, and compare SuCo with the baselines below.
>
>
> ||1.5B|||7B|||
> |----------|----|----|-----|----|----|-----|
> |Method|SFT|RL|Total|SFT|RL|Total|
> |Full CoT SFT|12|0|12|35|0|35|
> |AdaCoT|10|22|32|30|80|110|
> |AdaptThink|-|15|15|0|65|65|
> |S-GRPO|-|12|12|0|60|60|
> |LHRMs|8|15|23|25|65|90|
> |SuCo|5|17|22|20|70|90|
>
> Additionally, SuCo requires extra preprocessing time for MSC construction:
>
> | Total                     | 24   |
> | ------------------------- | ---- |
> | Identify raw MSC prefixes | 20   |
> | MSC refinement            | 4    |
>
>
> Although SuCo introduces a modest additional preprocessing cost, it achieves substantial reduction in inference tokens.

---

> > ### Author Rebuttal · Reviewer_fBsN · 2026-04-03
> >
> > Thank you for your detailed response. I think most of my concerns have been addressed. Overall, I'll be maintaining my positive score.

---

> > > ### Author Response · Authors · 2026-04-04
> > >
> > > Thank the reviewer for the thorough review and constructive suggestions. We are glad that our responses have addressed all your concerns.

---

### Decision · Program_Chairs · 2026-04-30

**Decision:**

Accept (regular)

**Comment:**

This paper proposes **SuCo**, a two-stage framework for adaptive reasoning control based on the notion of **Minimal Sufficient CoT (MSC)**, defined as the shortest reasoning prefix sufficient to produce the correct answer. The method combines MSC-aligned fine-tuning (MFT) with a reinforcement learning stage, Sufficiency-Aware Policy Optimization (SAPO), to encourage models to allocate reasoning effort continuously according to problem difficulty. The paper addresses an important and timely problem, namely reducing unnecessary chain-of-thought verbosity while preserving or improving task performance.

The submission has several strengths. Reviewers generally found the MSC formulation interesting and the overall framework well motivated. The empirical study is broad, covering math, code, and science benchmarks, and the rebuttal further strengthened the paper with additional analyses on calibration robustness, training cost, pass@k, the role of the refinement model, and out-of-domain generalization. In particular, the additional experiments suggest that the gains are not solely due to the 80B refinement model, that MSC-style supervision transfers across calibrator models, and that SuCo can generalize beyond the training domains better than the full-CoT baseline and other adaptive baselines.

At the same time, some limitations remain. The full pipeline is relatively complex, and parts of the method rely on strong auxiliary components, including MSC construction based on ground-truth answers and a powerful model for refinement. This constrains the most direct applicability to tasks with verifiable answers, and the conceptual and practical dependence on such components should be acknowledged clearly. Some reviewers also noted that the presentation could be clearer in places, and that the importance of the SAPO stage versus the already-strong MFT stage was not initially obvious, though the rebuttal provided useful additional evidence on this point. Overall, these concerns appear to be limitations in scope, clarity, and practicality rather than fatal flaws in the core empirical contribution.

On balance, I find this to be a solid paper with meaningful empirical contributions to adaptive reasoning efficiency. The core idea is interesting, the results are consistently strong, and the rebuttal addressed the major concerns in a substantive way. That said, I view the contribution as somewhat constrained by its reliance on verifiable-answer settings and by the complexity of the overall pipeline, so I do not see it as a clear accept at a higher confidence level. I therefore recommend **weak accept**. In the final version, the authors should more explicitly discuss the method’s scope and limitations, clarify the role of auxiliary components and SAPO, and improve presentation where the current exposition is difficult to follow.